# The global RNA-binding protein RbpB is a regulator of polysaccharide utilization in *Bacteroides thetaiotaomicron*

Ann-Sophie Rüttiger [1,2], Daniel Ryan [1,2], Luisella Spiga[3], Vanessa Lamm-Schmidt[2,4], Gianluca Prezza[2], Sarah Reichardt[2], Madison Langford[3], Lars Barquist [2,5,6], Franziska Faber [2,7], Wenhan Zhu [3] & Alexander J. Westermann [1,2] ✉

Paramount to human health, symbiotic bacteria in the gastrointestinal tract rely on the breakdown of complex polysaccharides to thrive in this sugar-deprived environment. Gut *Bacteroides* are metabolic generalists and deploy dozens of polysaccharide utilization loci (PULs) to forage diverse dietary and host-derived glycans. The expression of the multi-protein PUL complexes is tightly regulated at the transcriptional level. However, how PULs are orchestrated at translational level in response to the fluctuating levels of their cognate substrates is unknown. Here, we identify the RNA-binding protein RbpB and a family of noncoding RNAs as key players in post-transcriptional PUL regulation. We demonstrate that RbpB interacts with numerous cellular transcripts, including a paralogous noncoding RNA family comprised of 14 members, the FopS (family of paralogous sRNAs). Through a series of in-vitro and in-vivo assays, we reveal that FopS sRNAs repress the translation of SusC-like glycan transporters when substrates are limited—an effect antagonized by RbpB. Ablation of RbpB in *Bacteroides thetaiotaomicron* compromises colonization in the mouse gut in a diet-dependent manner. Together, this study adds to our understanding of RNA-coordinated metabolic control as an important factor contributing to the in-vivo fitness of predominant microbiota species in dynamic nutrient landscapes.

The obligate anaerobic, Gram-negative *Bacteroidales* represent a dominant order of the gut microbiota[1,2] and influence human health in various ways[3]. An important factor contributing to the success of these bacteria in colonizing the mammalian bowel is their immense metabolic capacities encoded on distinct polysaccharide utilization loci (PULs)[4]. PULs enable the catabolism of complex dietary polysaccharides and host mucus-derived glycans[5]. Each PUL encodes sets of membrane-spanning, glycolytic multi-protein complexes, which typically bind the respective substrate at the cell surface and cleave it into oligosaccharides. These oligosaccharides are imported into the periplasm through TonB-dependent transporters of the SusC family[6], and further processed into simple sugars. As individual PULs are substrate-specific, their expression is tightly controlled and responds to the nutritional fluctuations that are commonly associated with the

[1]Department of Microbiology, Biocenter, University of Würzburg, Würzburg D-97074, Germany. [2]Helmholtz Institute for RNA-based Infection Research (HIRI), Helmholtz Centre for Infection Research (HZI), Würzburg D-97080, Germany. [3]Department of Pathology, Microbiology, and Immunology, Vanderbilt University, Nashville, Tennessee, USA. [4]Institute of Molecular Infection Biology, University of Würzburg, Würzburg D-97080, Germany. [5]Faculty of Medicine, University of Würzburg, Würzburg D-97080, Germany. [6]Department of Biology, University of Toronto, Mississauga, L5L 1C6 Ontario, Canada. [7]Institute for Hygiene and Microbiology, University of Würzburg, Würzburg D-97080, Germany. ✉e-mail: alexander.westermann@uni-wuerzburg.de

dynamic gut environment. Therefore, PULs often include dedicated regulatory factors that spur transcription of their specific PUL operons when the corresponding carbon sources are sensed. This includes SusR-like regulators[7] and hybrid two-component systems[8,9] that combine both sugar-sensing and gene regulatory functions in a single polypeptide. Alternatively, transcription of PUL systems specific to the processing of host-derived glycans is typically governed by extra-cytoplasmic function sigma/anti-sigma factor pairs[10]. However, if and how *Bacteroidales* prevent ongoing translation of pre-existing PUL mRNAs once the inducing stimulus fades, is not currently known.

Only recently did RNA-seq studies from us[11,12] and others[13,14] map the transcriptome of select *Bacteroidales* members at single-nucleotide resolution. This entailed the identification of hundreds of noncoding RNA candidates, including the small noncoding RNAs (sRNAs) DonS and GibS. DonS is divergently encoded to an *N*-glycan-specific PUL and was shown to repress the expression of this PUL in *Bacteroides fragilis*[13], albeit through an unknown mechanism. The conserved sRNA GibS is induced in the presence of *N*-glycans, and binds and represses a glycoside hydrolase-encoding mRNA in *Bacteroides thetaiotaomicron*[11]. Given these examples and the general importance of RNA-based regulation, it is likely that *Bacteroides* spp. employ much more complex sRNA-based post-transcriptional regulatory networks to optimize fitness in response to dynamic nutrient levels. However, the identification of post-transcriptional regulatory circuits in these bacterial taxa that lack homologs of classical RNA chaperones has proven challenging.

The present study reports the identification of an in-vivo phenotype, the RNA interactome, a metabolism-associated function, and the underlying molecular mechanism of the first global RNA-binding protein (RBP) in *Bacteroides*. Specifically, we found that a *B. thetaiotaomicron* mutant devoid of the RNA recognition motif 1 (RRM-1)-containing protein RbpB[15,16] failed to efficiently colonize the mammalian intestine in a diet-dependent manner. Cross-linking immunoprecipitation and sequencing (CLIP-seq) of *B. thetaiotaomicron* RbpB demonstrated that this protein is a global RNA binder. Analysis of the RbpB interactome led to the discovery of a complex RNA network, with a highly conserved multicopy sRNA family at its center: the 'FopS' cluster (for 'family of paralogous sRNAs'). While the 14 FopS members of *B. thetaiotaomicron* exhibit partial sequence similarity to the previously characterized GibS sRNA[11], we provide evidence of functional diversification between GibS and FopS. The FopS sRNAs function as post-transcriptional repressors of specific PUL operons by binding to the vicinity of the start codon of the first mRNA of target PUL polycistrons, typically corresponding to the cognate SusC homolog, impeding translation of the respective glycan transporter. As demonstrated for the high-mannose *N*-glycan-catabolizing PUL72, RbpB counteracts FopS activity, relieving translational repression. Together, this study reports a remarkably complex post-transcriptional control network that allows *Bacteroides* to switch off specific PULs in the absence of the cognate substrate or in the presence of an alternative, preferred carbohydrate source to optimize fitness.

## Results

### RbpB promotes mucus-foraging by *B. thetaiotaomicron* in the mammalian intestine

*Bacteroides* spp. encode an array of transcription factors that govern carbohydrate utilization in these gut bacteria[9,10,17–21]. Conversely, there is currently little knowledge as to the extent to which post-transcriptional control impacts on *Bacteroides* metabolic competitiveness. Given that the deletion of the RbpB-encoding gene (*BT_1887*) in *B. thetaiotaomicron* led to the differential expression of several PUL genes[16], we assessed the fitness of a *B. thetaiotaomicron* Δ*rbpB* mutant in the mammalian gut in response to distinct diets. As conventionally raised mice are resistant to *B. thetaiotaomicron* colonization[22], we treated C57BL/6 mice with an antibiotic cocktail to promote *B.*

*thetaiotaomicron* engraftment. We then inoculated mice fed a conventional diet with an equal ratio of the *B. thetaiotaomicron* wild-type strain and an isogenic Δ*rbpB* mutant, followed by determination of the abundance of each strain in cecal and colonic contents 6 days post-inoculation by plating on selective media (Fig. 1a).

Notably, the wild-type was outcompeted by the Δ*rbpB* mutant in the fiber-rich control diet (black dots in Fig. 1b). Remarkably, however, the Δ*rbpB* mutant displayed a significant fitness disadvantage compared to the wild-type strain when switched to a low fiber diet, which drives the bacteria to feed on host-derived glycans (red dots in Fig. 1b). Consistent with the notion that the colon has a thicker mucous layer than the cecum, the fitness defect was more pronounced in the colon. This phenotype was specific to RbpB as *trans*-complementation of the gene into the Δ*rbpB* background reverted the in-vivo fitness to that of the wild-type strain (triangles in Fig. 1b). Moreover, we observed a sex-specific difference, in which the fitness defect of the Δ*rbpB* mutant was more prominent in males than in females (Fig. 1c). The magnitude of this attenuation exceeded the relatively mild in-vivo phenotypes associated with individual PUL-encoded transcriptional regulators (e.g. refs. [23,24]) and rivaled the large impact of deleting the transcriptional master regulator of polysaccharide utilization, Cur[25].

A 16S ribosomal RNA gene analysis of the fecal samples at the end of the experiment revealed significant differences in the resident microbiota composition between the diet groups in terms of taxa (Supplementary Fig. 1a), beta diversity (Supplementary Fig. 1b, c), and alpha diversity (Supplementary Fig. 1d, e). While we cannot rule out the influence of microbe-microbe interactions on *B. thetaiotaomicron* colonization, the competitive setup ensured the two experimentally introduced strains to experience the same nutritional environment and interactions with other members of the microbiota within individual mice. The difference in relative fitness, therefore, suggests a role of RbpB in promoting the mucus-foraging lifestyle of *B. thetaiotaomicron*, prompting us to investigate the cellular function of this protein in more detail.

### RbpB acts as a global RNA-binding protein in *B. thetaiotaomicron*

RbpB was reported to possess the ability to bind synthetic single-stranded RNA 'pentaprobes' in vitro[16]. However, whether *Bacteroides* RbpB acts as a global RNA binder in vivo—as opposed to a test tube—was previously not assessed. To explore the in-vivo functions of RbpB, we constructed a C-terminally FLAG-tagged version of RbpB and stably integrated it into the chromosome of *B. thetaiotaomicron*. In-vivo crosslinking and immunoprecipitation (CLIP) experiments (Supplementary Fig. 2a) revealed the characteristic crosslink-induced signal on an autoradiograph (Fig. 2a), which was sensitive to RNase I—but not to DNase I—treatment (Supplementary Fig. 2b), suggesting the protein interacts primarily with cellular RNA.

To map the RbpB interactome at genomic scale, we subjected the co-purified RNA to high-throughput sequencing (CLIP-seq). Two independently performed replicate experiments showed clear read enrichments in crosslinked samples as compared to matched background controls (Supplementary Fig. 2c, d). We report a total of 213 significantly enriched ($p_{adj} \leq 0.05$; $\log_2$ FC > 2), manually confirmed peaks within 173 different mRNAs (12 peaks in 5′ UTRs, 175 in CDSs, 26 in 3′ UTRs), 64 peaks in 57 different sRNAs (two of which overlap with 3′ UTR peaks and three with 5′ UTR peaks, and one or two peaks, respectively, map to the 6S and 4.5S housekeeping RNAs), and 14 peaks within intergenic regions (Fig. 2b, c; Supplementary Data 1). We integrated the RbpB binding site annotation in the open-access online resource 'Theta-Base' (https://bacteroides.helmholtz-hzi.de/jbrowse_new/)[12] to facilitate interrogation by the public.

Inspecting the peak size distribution revealed two over-represented footprint lengths (Fig. 2d), reflecting the band pattern on the autoradiogram (Fig. 2a; Supplementary Fig. 2b). While a shorter

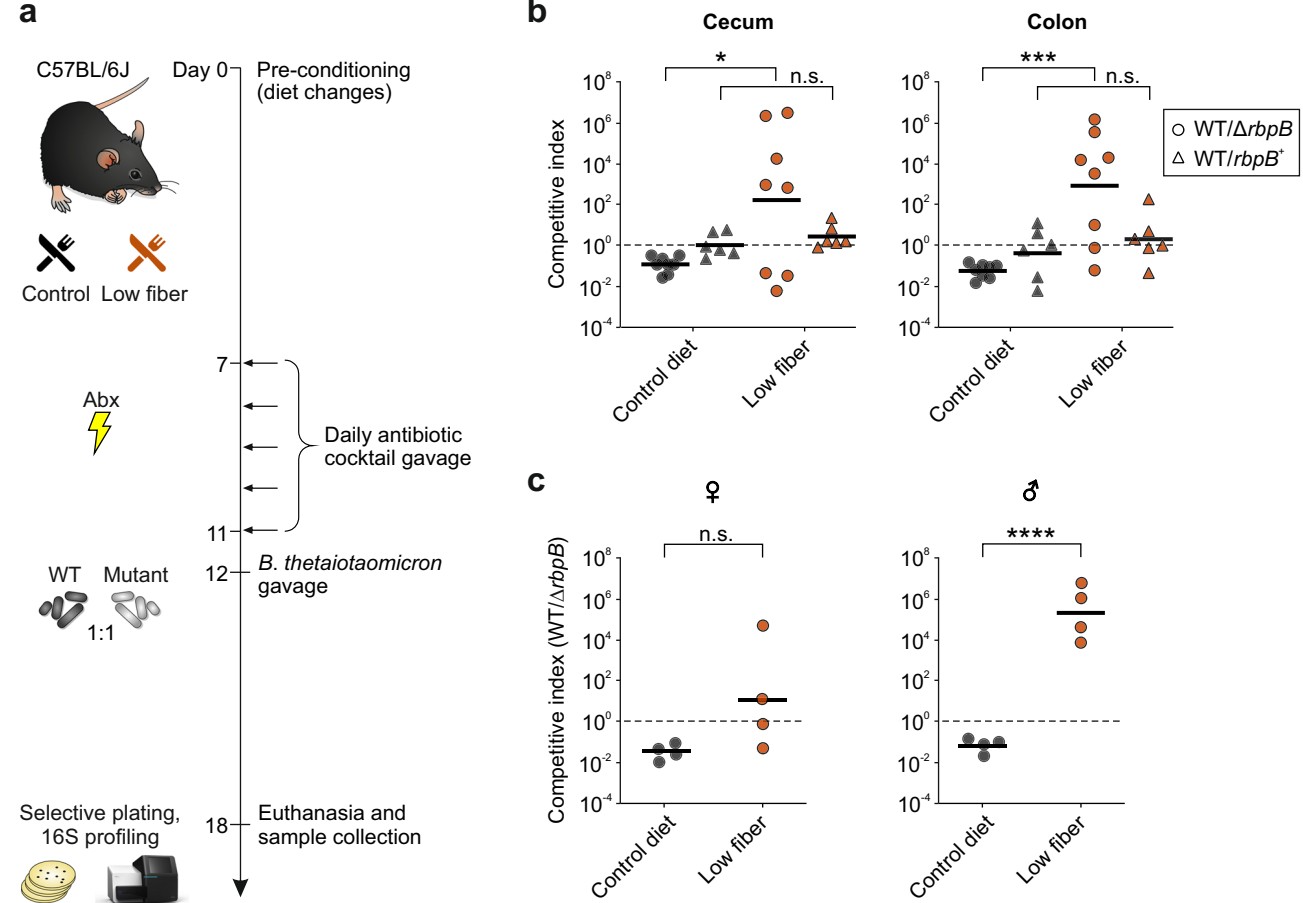

**Fig. 1 | RbpB promotes mucus-foraging in the mammalian intestine.**
**a** Schematic representation of the experimental outline. One week before antibiotic treatment, C57BL/6J mice were switched to a low fiber diet or remained on the fiber-rich control diet until the end of the experiment. Antibiotic cocktails were administered by oral gavage daily for 5 days. Mice were inoculated with an equal mixture of $0.5 \times 10^9$ CFU of the *B. thetaiotaomicron* wild-type and *rbpB* mutant strains. After 6 days, cecal and colonic tissue was collected and the bacterial numbers determined by plating on selective agar. **b** The competitive index of *B. thetaiotaomicron* wild-type versus Δ*rbpB* or *rbpB*+ in the cecal (left) or colonic (right) contents. The data are combined over two independent experiments, each comprised of four biological replicates (three biological replicates for WT/*rbpB*+ competition). Black horizontal lines represent the geometric means. *$p = 0.0245$, degrees of freedom (df) = 14, 95% confidence interval (CI): 1.169–14.56; ***$p = 0.0007$, df = 14, 95% CI: 5.268-15.67. **c** Same colonic data as in (**b**), separated by sex (female mice on the left, and male mice on the right). ****$p < 0.0001$, df = 6, 95% CI: 11.14–18.89; n.s., not statically significant based on two-tailed Student's *t*-tests. Source data are provided as a Source Data file.

peak size of ~30 nt was predominantly associated with mRNAs and intergenic regions, peaks within sRNAs exceeded 50 nt (Fig. 2d). Functional analysis of RbpB-bound mRNAs revealed an enrichment of peaks in transcripts encoding proteins involved in translational processes, such as the biosynthesis of aminoacyl-tRNAs and ribosomal proteins (Supplementary Fig. 2e), implying a translation-related role of this RBP.

In Gram-negative species, global RBPs often stabilize their RNA ligands by shielding them from cellular nucleases[26–29]. To test for a similar role of *Bacteroides* RbpB, we coupled rifampicin-mediated inhibition of de-novo transcription to the measurement of RNA decay kinetics[30]. This analysis suggested RbpB overexpression (Supplementary Fig. 3a) to reduce the cellular half-lives of some of the top-enriched mRNA ligands, whereas its deletion entailed more subtle effects (Supplementary Fig. 3b). Closer inspection of CLIP-seq peaks suggested the observed mRNA destabilizations upon RbpB over-expression to be independent of the relative position of the binding sites within those transcripts (Supplementary Fig. 3d). RbpB-associated sRNAs were generally very stable, regardless of the presence or absence of RbpB (Supplementary Fig. 3c). It therefore appears that the primary role of RbpB is not to protect its RNA ligands from degradation.

With respect to RbpB-bound noncoding RNAs, drawing from available secondary structure information[15], manual inspection revealed the protein to bind preferentially within single-stranded regions (Supplementary Fig. 2f). MEME analysis[31] identified several primary sequence motifs enriched in RbpB peaks (Supplementary Fig. 4a), with the most significant comprising a 41 nt-long motif (Fig. 2e). In fact, many of the most strongly enriched sRNA ligands of RbpB contained this sequence around the RBP footprint (Supplementary Fig. 4b). Given this observation, we focused on this sequence motif and the sRNAs containing it.

### A conserved multicopy sRNA family is associated with RbpB
We next set out to determine the prevalence of the above sequence motif. A hidden Markov model-based iterative sequence homology search returned 14 paralogous sequences within the *B. thetaiotaomicron* VPI-5482 genome (Fig. 3a). They all fell within annotated sRNAs, which were significantly enriched in the RbpB CLIP-seq dataset. A BLAST search revealed this multicopy sRNA family to be highly conserved within *Bacteroidaceae* (Supplementary Fig. 5a), ranging from 12 copies (in *Bacteroides caccae* and *Bacteroides uniformis*) to 15 copies per genome (in *Bacteroides xylanisolvens*) (Supplementary Fig. 5b). Given the presence of the sequence in a Caudovirales phage

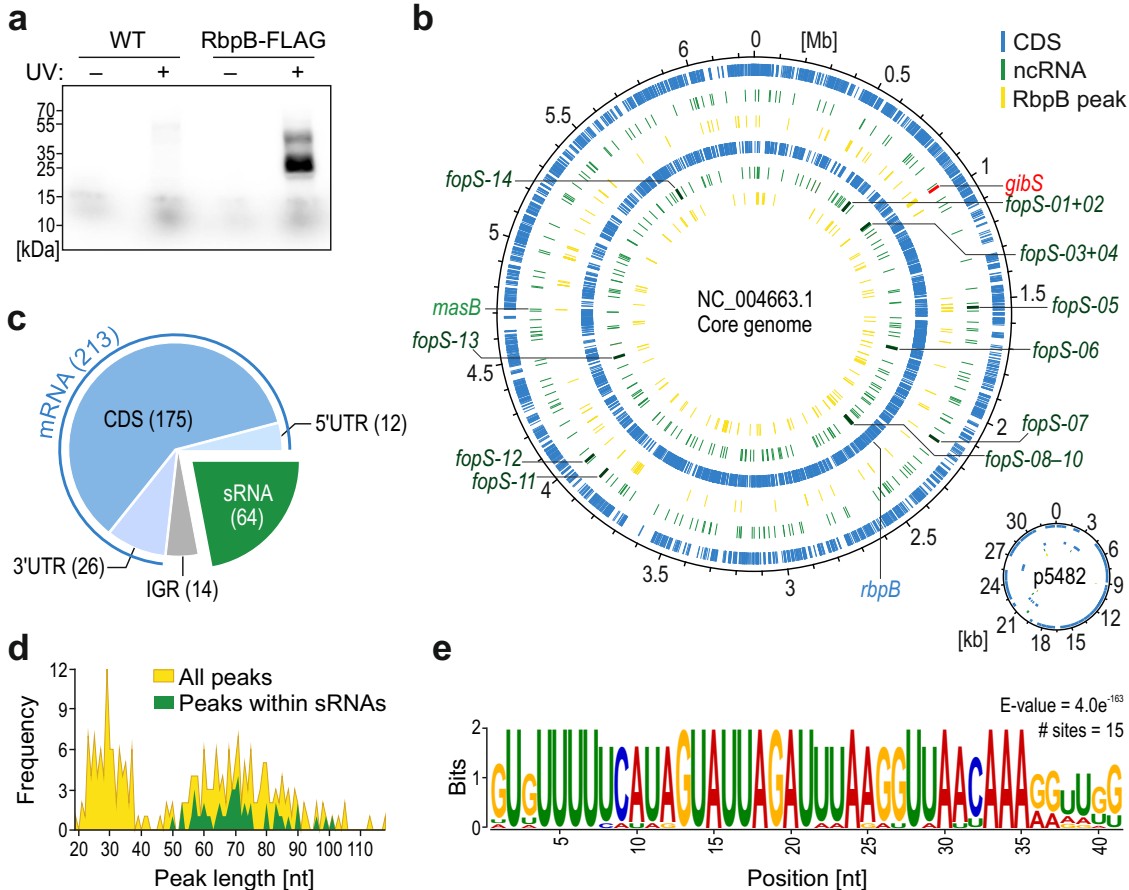

**Fig. 2 | RbpB acts as a global RNA-binding protein in *B. thetaiotaomicron*.**
**a** Autoradiogram of radioactively labeled RNAs covalently bound to RbpB-3xFLAG after in-vivo UV crosslinking, immunoprecipitation, and transfer to a nitrocellulose membrane. Shown is one representative image of two independent replicate experiments. **b** Genomic positions of the identified RbpB CLIP peaks in the context of the *B. thetaiotaomicron* chromosome and plasmid. The positions of coding sequences (CDS) and noncoding RNA (ncRNA) genes were retrieved from NCBI and published literature[11]. The *gibS* gene and the 14 FopS-encoding loci (which will become relevant below) are marked. The *rbpB* mRNA itself harbors a RbpB binding site, implying the potential for autoregulation. Inner circles refer to the minus strand and outer circles to the plus strand. **c** RNA class distribution of RbpB ligands. Numbers in brackets refer to statistically significant and manually confirmed peaks supported by two independent CLIP-seq experiments. Note that the sum of peaks mapping to the different genetic features exceeds the actual total peak number as some peaks mapped to overlapping annotations (such as 3′ UTRs and 3′-derived sRNAs). **d** Peak size distribution. Plotted are the frequencies for individual peak sizes of all unique peaks (yellow) and sRNA peaks (green). **e** Enriched sequence motif within RbpB sRNA ligands. Source data are provided as a Source Data file.

(Supplementary Fig. 5a)—predicted to target *Bacteroides* based on matching CRISPR spacers[32]—bacteriophages may have played a role in disseminating the corresponding sRNAs within and across *Bacteroidaceae* genomes. We term the cluster "family of paralogous sRNAs" (FopS).

The GibS sRNA was also pulled down together with RbpB (Fig. 2b). Previously, we had shown that this sRNA comprises two distinct seed regions, referred to as R1 and R2, which mediate basepairing with—and repression of—its two direct target mRNAs[11]. Interestingly, the FopS family members contain the R1 sequence of GibS but lack R2 (Fig. 3a). We adopted a structural RNA alignment tool[33] to infer the consensus secondary structure of the 14 FopS sRNAs of *B. thetaiotaomicron*. This structure comprised a single-stranded 5′ stretch followed by a Rho-independent terminator hairpin (Fig. 3b), and was deposited in the Rfam database[34]. The conserved R1 sequence is located within the predicted single-stranded region at the 5′ end of the FopS consensus structure, as would be expected if R1 acts as a seed sequence in these sRNAs.

Electrophoretic mobility shift assay (EMSA) with recombinant *B. thetaiotaomicron* RbpB (Supplementary Fig. 6a) and—as a representative sRNA family member—in-vitro-transcribed FopS-10 supported the formation of stable ribonucleoprotein (RNP) particles (Fig. 3c; Supplementary Fig. 6b). Inverting the sequence of the 55 nt-long RbpB

binding site (as deduced from CLIP-seq) predicted to largely maintain the secondary structure of FopS-10 (Supplementary Fig. 6c) impeded the formation of RNP complexes ('Inv' in Fig. 3c; Supplementary Fig. 6b). EMSAs also validated that RbpB binds to GibS (Supplementary Fig. 6d). In the case of both tested sRNAs, the affinity of the protein to its ligand was in the low micromolar range, and thus an order of magnitude lower than what is typically observed in interactions of pseudomonadotal Hfq and ProQ with their cognate sRNA partners[35–37]. Taken altogether, these data support the CLIP-seq results and confirm that RbpB binds to FopS and GibS sRNAs in a sequence-specific and concentration-dependent manner.

## FopS expression responds to bile salts

Oftentimes, expression profiling of a given sRNA provides a glimpse into its function. While GibS transcription is driven from a non-canonical promoter associated with stationary phase-induced genes[11], inspection of the regions upstream of the *fopS* genes revealed the presence of the canonical σ^ABfr promoter[38] (red or green boxes, respectively, in Fig. 3a). In accordance with an independent transcriptional activation, published RNA-seq data[12] showed an anticorrelated expression pattern of GibS and the FopS sRNAs (Fig. 3d). As confirmed by northern blotting (Fig. 3e), the steady-state level of GibS increased during growth in rich TYG medium up to stationary phase,

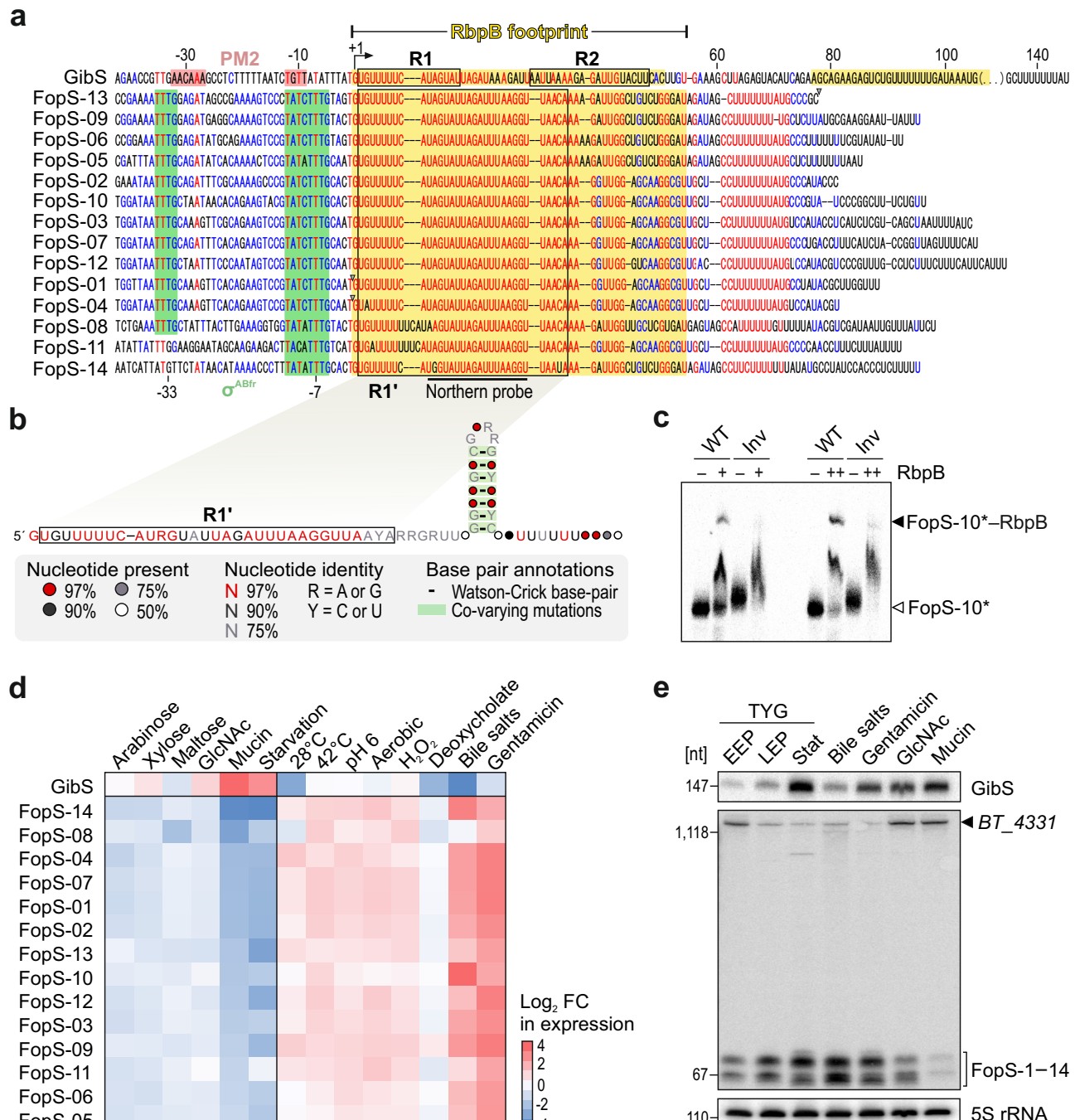

**Fig. 3 | A fourteen-member family of paralogous sRNAs associates with RbpB.**
**a** Sequence alignment of R1-containing sRNA genes from *B. thetaiotaomicron* VPI-5284. Red and blue letters indicate highly conserved and less-conserved ribonucleobases, respectively. Numbers denote the positions relative to the 5′ end of GibS (bent arrow). Canonical σ[ABfr] promoters[38] (green boxes), promoter motif 2[11] ('PM2'; pale red boxes), and the R1 and R2 seeds (as determined in ref. [11]) are indicated. R1′ refers to the 3′-extended seed of FopS sRNAs (see Fig. 4e). Curated sRNA boundaries are marked by gray triangles (see Methods for details). Sequence identity of the 14 individual FopS paralogs with GibS ranges between 28.4-43.9%. **b** FopS consensus structure predicted using the Webserver for Aligning structural RNAs[33]. Secondary structure was visualized using the R2R software[93]. R1′ sequence is boxed. **c** In vitro-transcribed, radioactively labeled FopS-10 was incubated for 1 h with two defined concentrations of recombinant RbpB ('+': 10 μM; '++': 20 μM) and the

resulting complexes resolved on denaturing gels. 'WT' refers to the wild-type sRNA sequence and 'Inv' to a FopS-10 variant with an inverted RbpB-target sequence (see Supplementary Fig. 6c). **d**, **e** Expression profiling of GibS and FopS sRNAs. Relative expression of the indicated sRNAs over a range of carbon sources (relative to growth in minimal glucose-containing medium) and stress conditions (relative to an unstressed control) based on RNA-seq data from[12] (**d**) and northern blot-based validation (**e**). EEP, early exponential phase; MEP, mid-exponential phase; stat, stationary phase in TYG. The annealing site of the FopS-specific northern probe is indicated in panel **a**. The upper band corresponds to the primary transcript of 3′ UTR-derived FopS-14. Positions of the marker bands are shown to the left. 5S rRNA served as loading control. Images representative of two (**c**) and three (**e**) independent replicates, respectively. Source data are provided as a Source Data file.

but was highest when bacteria fed on *N*-Acetylglucosamine (GlcNAc)-containing carbohydrates (as previously reported[11]), particularly when mucin was the sole carbon source. In contrast, GibS levels dropped when bacteria were exposed to bile salts. Under those conditions, *B. thetaiotaomicron* induced expression of the FopS sRNAs, as reflected by two, relatively distinct bands that matched the predicted length of the FopS sRNAs (two clusters of ~65 and ~70 nt) on a northern blot (Fig. 3e) using a probe against the universal FopS 5′ region (Fig. 3a). An additional, high molecular band of ~1,400 nt was most likely derived from *BT_4331*, i.e. the parental transcript of the only 3′ end-derived member of this sRNA family, FopS-14 (Supplementary Fig. 6e). Concentration range experiments coupled to northern blotting revealed FopS expression to peak between 0.03 and 0.1 mg/mL of bile salts (Supplementary Fig. 6f, g), falling within the range of their in-vivo concentrations in the human large intestine[39,40]. Responsiveness to this in-vivo-relevant stimulus corroborates the relevance of the identified sRNA-RBP cluster for *B. thetaiotaomicron* within its host niche.

## Established GibS targets are refractory to FopS-mediated regulation

We next sought to functionally characterize the FopS sRNA family. Previously we found that GibS represses *BT_0771* and *BT_3893*, which code for a glucan-branching enzyme that belongs to the glycoside hydrolase 13 family (http://www.cazy.org/) and a hypothetical protein, respectively[11]. Mechanistically, GibS binds the translation initiation regions of *BT_0771* and *BT_3893* mRNAs, the latter of which by engaging both the R1 and R2 seeds[11]. To assess if these GibS targets are also regulated by the FopS cluster, we compared their steady-state transcript levels between wild-type and three independent *fopS* deletion mutants (Δ*fopS-09*, Δ*fopS-10*, Δ*fopS-13*; i.e., sRNAs, whose 5′ end matches the FopS consensus, yet reflect downstream sequence diversity at positions 35-60). In stationary phase TYG cultures, endogenous GibS and FopS sRNAs were relatively highly expressed (Fig. 3e). However, both established GibS target mRNAs were derepressed only in the absence of GibS and not in the absence of individual FopS sRNAs (Fig. 4a). Consistent with the notion that GibS is the primary regulator of these mRNAs, in-vitro-transcribed 5′ ends of *BT_3893* and *BT_0771* mRNAs annealed with radiolabeled GibS, but not (*BT_3893*) or less efficiently (*BT_0771*) with FopS-10 (Supplementary Fig. 7a). Therefore, although functional redundancy amongst the paralogous sRNAs might partially compensate for effects derived from single *fopS* deletion, it appears from EMSAs that FopS sRNAs are relatively ineffective in regulating GibS targets, despite possessing one of the two seed regions of GibS.

## FopS sRNAs repress PUL72 with phenotypic consequences during growth on high-mannose *N*-glycans

To identify the bona fide function of the FopS's, we applied the IntaRNA algorithm[41] using the consensus FopS 5′ sequence (position 1 to 21) as query (Supplementary Data 2). The proposed target candidates were strongly enriched in mRNAs encoding outer membrane proteins, particularly SusC-like transporters of PUL systems. Interrogation of available RNA-seq data[12] revealed a strong anticorrelation between the expression of individual FopS sRNAs and that of their in-silico-predicted, common targets (Fig. 4b), implying negative regulation to prevail amongst putative FopS-mediated activities.

For further characterization of FopS-mediated target control, we focused on PUL72, a functionally and structurally characterized PUL specific for mammalian high-mannose-type *N*-glycans[20,42,43]. IntaRNA predictions were corroborated by the steady-state transcript levels of *BT_3983*, encoding the SusC homolog of this PUL. In line with a repressive effect of the FopS sRNAs on this SusC-like transporter, its mRNA level was elevated in individual Δ*fopS* strains during growth under bile stress, and returned to basal level when the respective *fopS* was complemented in *trans* (Fig. 4c). The genes of the PUL72-encoded surface glycan-binding SusD homolog (*BT_3984*) and, to lesser extent,

of *BT_3985-88* (glycoside hydrolase genes) that are co-transcribed with the *susC* homolog, showed a similar FopS-dependent expression profile (Fig. 4c; Supplementary Fig. 7b), suggesting that FopS binding to the 5′ region of *susC^{BT_3983}* also represses the downstream genes in the corresponding operons. The expression of other PUL72 genes, encoded from the opposing DNA strand, was hardly altered by FopS depletion and deletion of *gibS* did not affect the steady-state levels of any of these transcripts (Supplementary Fig. 7b).

Intending to inactivate multiple FopS family members simultaneously, we generated a ΔΔ*fopS-09/10* double-knockout mutant. However, attempts to delete additional *fopS* genes in this strain background failed after repeated attempts. We therefore opted for a CRISPR interference (CRISPRi)-based approach[44] to transcriptionally silence the remaining family members. Evaluating the knockdown efficiencies of three different guide RNAs (gRNAs) targeting the universal sequence at the FopS 5′ end (Supplementary Fig. 8a), we identified one strain (expressing gRNA #3; termed "ΔΔ+gRNA3") in which FopS aggregate levels reduced to ~30% upon inducing the CRISPRi system (Supplementary Fig. 8b, c). When grown in minimal medium containing the PUL72 substrate high-mannose *N*-glycan as the sole carbon source, FopS expression was ~3-fold reduced as compared to growth on glucose (Fig. 4d, right; Supplementary Fig. 8e). Importantly, the ΔΔ+gRNA3 strain exhibited enhanced growth on this carbon source relative to a wild-type culture (Fig. 4d, left). This is likely a result of elevated PUL72 activity in the mutant background, as the transcript levels of the PUL72-associated SusCD homologs increased two- to three-fold in the ΔΔ+gRNA3 compared to the wild-type strain (Fig. 4d, right; Supplementary Fig. 8f). Collectively, these data therefore suggest that the FopS sRNAs—but not GibS—repress PUL72 and this regulation entails phenotypic consequences when *B. thetaiotaomicron* consumes high-mannose *N*-glycans.

## A panel of PUL polycistrons are targeted by FopS at their 5′ end

In-silico prediction of RNA-RNA interaction suggested that the FopS 5′ end basepaired with the translation initiation region of *susC^{BT_3983}* (Fig. 4e). This involves a 3′-extended version of the R1 region within FopS (that we term R1′)—a sequence that includes several mismatches in GibS (Fig. 3a), which might explain the divergent outcomes of deleting these sRNAs on levels of the *susC^{BT_3983}* mRNA (Fig. 4c). Indeed, inline probing of the radiolabeled 5′ region of a *susC^{BT_3983}* mRNA fragment (between the 5′ end and the 69th codon of the coding sequence) upon incubation with increasing concentrations of in-vitro-transcribed GibS or FopS-10 showed the predicted targeting site is shielded by FopS-10, but not by GibS (Fig. 4f; compare lanes 10 and 11, or 6 and 7, respectively, with lane 4). EMSA likewise confirmed binding of FopS-10 to the 5′ region of *susC^{BT_3983}* mRNA, whereas the addition of up to 1 μM of GibS did not result in an upshift of the radioactively labeled mRNA fragment (Supplementary Fig. 7c). Point-mutating eight nucleotides within the predicted FopS-10 targeting region of *susC^{BT_3983}* mRNA ('Mut') fully abrogated sRNA binding in vitro and introduction of the corresponding compensatory mutations into FopS-10 restored binding to the mutant *susC^{BT_3983}* variant (Supplementary Fig. 7d), thus supporting the in-silico-predicted interaction site (Fig. 4e).

Finally, EMSAs confirmed the predicted interaction of FopS-10 with eight additional target candidates in vitro, including the 5′ regions of mRNAs of five additional SusC homologs from various PULs (PUL12, -51, -77, -80, -88) and the mRNAs of three glycoside hydrolases, two of which are also encoded within PULs (Supplementary Fig. 7e). These latter PULs (PUL20 and -44) exhibit an atypical genomic architecture, wherein the respective FopS target glycoside hydrolase (rather than the cognate SusC homolog) is encoded by the first operon gene[4,12]. Taken together, these data suggest the FopS sRNA family to repress multiple, mostly host glycan-catabolic[10] PUL systems by direct basepairing to the translation initiation region at the 5′ end of the respective polycistronic mRNA.

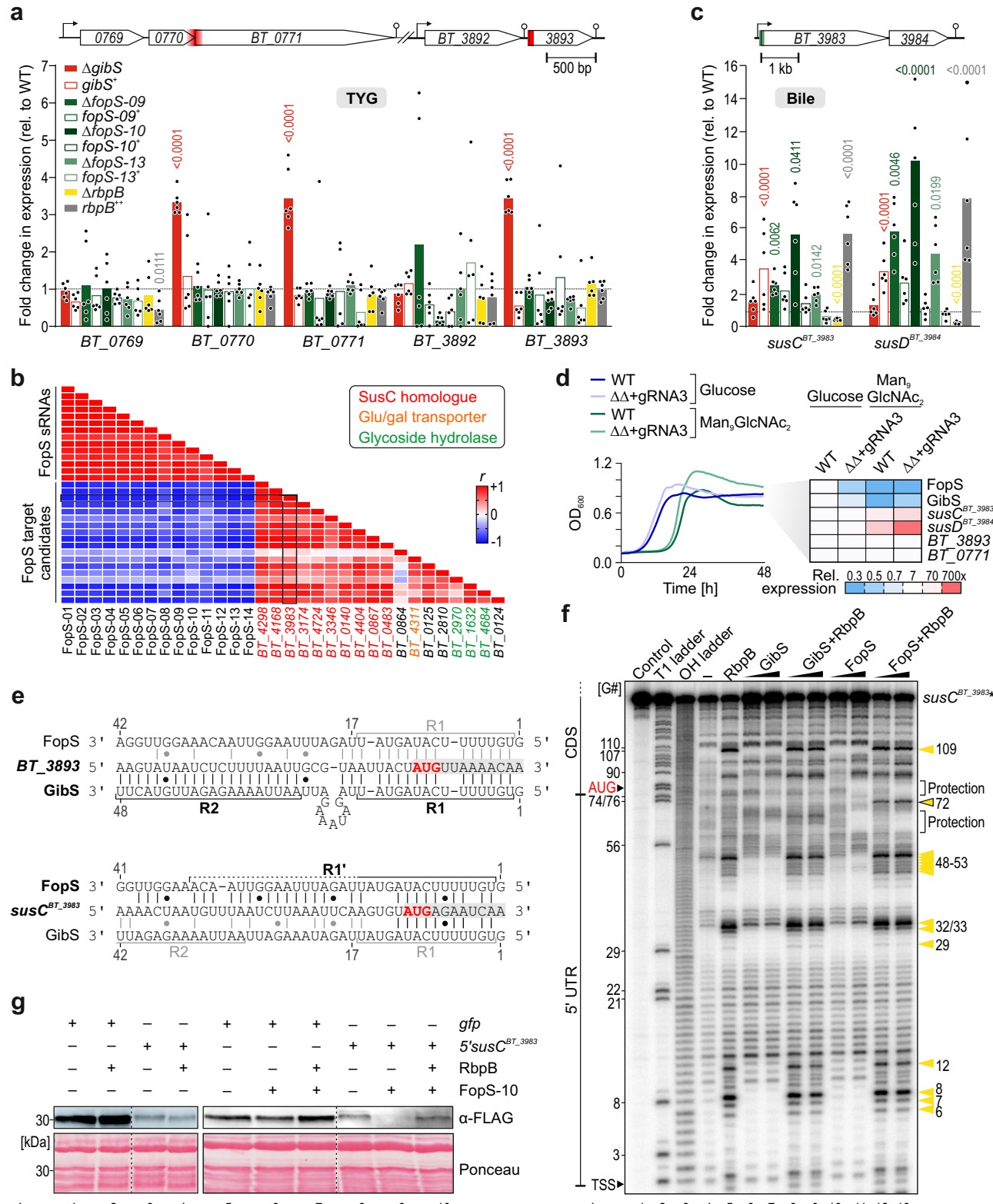

## FopS-10 suppresses translation initiation of *susC^BT_3983* and is antagonized by RbpB

Current paradigm holds that RNA-binding proteins can repress target mRNA translation by facilitating sRNA binding, in which the specificity of the binding is determined by the basepairing between the sRNA and the target. To investigate whether RbpB assumes similar roles, we performed three-component EMSAs. Compared to the affinities of GibS and FopS-10 to their respective targets in the absence of RbpB, we

(black curves in Supplementary Fig. 9a), supplementation of a fixed concentration (1 μM) of the protein did not affect sRNA-mRNA duplex formation efficiency in vitro (yellow curves in Supplementary Fig. 9a). Gradually increasing the RbpB concentration to up to 20 μM suggested the formation of a ternary complex consisting of GibS, its target mRNA (*BT_3893*), and RbpB, whereas the protein was not engaged in a stable ternary complex with FopS-10 and most of its mRNA targets (except for *BT_2970*) under the same conditions (Supplementary Fig. 9b, c, d).

**Fig. 4 | FopS and RbpB constitute a post-transcriptional layer of PUL control.**
**a** qRT-PCR analysis of direct GibS targets[11] in stationary phase in TYG. Bars denote the mean from six replicate measurements. Significance was assessed using two-way ANOVA (Sidak's multiple comparisons test; shown are significant changes [$p < 0.05$] relative to wild-type [dashed line]). Target locus representations at the top, with experimentally mapped GibS binding sites[11] in red. **b** Pearson's correlation coefficients ($r$) for individual sRNA-mRNA pairs based on RNA-seq data[12]. As an example, the (anti-)correlation of $BT\_3983$ with FopS and other predicted targets is boxed. **c** qRT-PCR measurement of $susCD^{BT\_3983-84}$, analogous to panel (**a**), except that strains were exposed for 2 h to 0.5 mg/mL of bile salts. Locus representation on top, with FopS binding site (determined in panels **e**, **f**) in green. **d** Left: growth kinetics of a multi-*fopS* mutant ('ΔΔ+gRNA3') and its parental wild-type on either glucose or Man$_9$GlcNAc$_2$ (mean of two biological replicates). Right: expression of selected sRNAs and mRNAs at 48 h of growth (relative to wild-type in glucose). For

underlying expression data, see Supplementary Fig. 8d-f. **e** RNA:RNA interaction predictions. Start codons are red and gray shadings denote coding regions. R1, R2: seed regions of GibS[11]; R1': 3'-extended seed of the FopS's. Coordinates relative to 5' ends. **f** In-line probing of 0.2 pmol $^{32}$P-labeled $susC^{BT\_3983}$ in absence or presence of either 20 nM or 200 nM GibS, FopS-10, or both sRNAs, and 20 μM of recombinant RbpB. 'Control' and '−' refer to the radiolabeled RNA substrate in water or inline buffer, respectively. Partially RNase T1- ('T1') or alkali-digested ('OH') substrates were included as ladders. **g** In-vitro translation of $5'\,susC^{BT\_3983}$-*gfp-3xFLAG* or *gfp-3xFLAG* (control). 1 μM (lanes 1–4) or 0.5 μM (lanes 5–10) of mRNA was in vitro-translated with reconstituted *E. coli* 70S ribosomes, with or without 20 μM purified RbpB and 25 μM in-vitro-transcribed FopS-10. Translation products were detected via western blotting and Ponceau staining served as loading control. Images representative of two (**f**) or three (**g**) independent replicates. Source data are provided as a Source Data file.

Instead, high concentrations of the protein ($> 5\,\mu M$) titrated FopS-10 away from its targets in vitro (Supplementary Fig. 9c, d). Likewise, overexpression of the protein in vivo resulted in increased, and *rbpB* deletion in reduced steady-state levels of the PUL72 *susCD* pair (Fig. 4c), suggesting that RbpB counteracts FopS-mediated target repression.

EMSAs also revealed that the $susC^{BT\_3983}$ and $BT\_3893$ mRNAs themselves are ligands of RbpB—even in the respective absence of FopS or GibS (Supplementary Fig. 9b, c). Note that none of these genes was expressed during mid-exponential growth in rich medium[11], providing a plausible explanation why the corresponding mRNAs were not recovered in the CLIP-seq experiment (Fig. 2). This sRNA-independent interaction was further supported by inline probing, revealing substantial structural rearrangements in the 5' UTR of $susC^{BT\_3983}$ in the presence of RbpB (compare lanes 4 and 5 in Fig. 4f). Inline probing of the interaction between RbpB and this 5' fragment in the additional presence of FopS-10 uncovered one further structural change (a band present only in lanes 12 and 13 in Fig. 4f). This rearrangement occurred at position 72 relative to the 5' end, which coincides with position −6 with respect to the start codon of the $susC^{BT\_3983}$ mRNA (see Supplementary Fig. 10a for a model of the structural rearrangements) and thus, falls within the critical window for efficient translation initiation in the *Bacteroidota*[45].

In-vitro translation of a $susC^{BT\_3983}$-*gfp-3xFLAG* mRNA template was unaffected by the addition of recombinant RbpB alone (Fig. 4g, lanes 4 vs. 3). In contrast, addition of FopS-10 to the reaction mix abrogated target protein synthesis (Fig. 4g, lanes 9 vs. 8). This effect was sequence-specific as evidenced by the point-mutated FopS-10 variant (see Supplementary Fig. 7d), which failed to repress the translation of wild-type $susC^{BT\_3983}$, yet effectively inhibited that of a template harboring the respective compensatory mutations (Supplementary Fig. 10b). In the additional presence of recombinant RbpB, FopS-10-mediated translational repression was relieved (Fig. 4g, lane 10). Translation of a control template, consisting of an *Escherichia coli* ribosome binding site fused to the 5' region of the GFP open reading frame and a triple-FLAG tag, was largely unchanged in analogously prepared reactions (Fig. 4g, lanes 1, 2, 5–7). Based on these data, we propose that FopS-10 sRNA blocks translation of the $susC^{BT\_3983}$ mRNA by binding adjacent to the start codon and providing steric hindrance to the initiating ribosome. This repression is counteracted by RbpB, possibly by titrating FopS-10 away from its target mRNA and/or by opening up the translational enhancer region of $susC^{BT\_3983}$ that is otherwise occluded by FopS-10 (Supplementary Fig. 10a). Altogether, our mechanistic data suggest that, in contrast to well-characterized RNA-binding proteins, RbpB assumes a role in excluding the interaction of sRNAs with a range of target transcripts, serving as a post-transcriptional hub in *Bacteroides* metabolic control.

## Discussion

The composition of the gut microbial community and, therefore, the pivotal functions these microbes provide to human health hinges on

their ability to persist in the face of nutritional fluctuation. Understanding how gut commensals adapt their metabolism to the daily variations in feeding rhythm and types of diet has thus become an important branch of microbiota research[46,47]. As one of the predominant bacterial genera in the healthy human gut microbiota, *Bacteroides* possess dozens of multi-protein complexes encoded on dedicated genomic loci—the PULs—to bind, clip, and import specific polysaccharides. Historically, PUL function was first studied in *B. thetaiotaomicron*[48,49], serving as a paradigm for polysaccharide breakdown in the human microbiome, with practical applications[50]. Complementing previous studies that focused on transcriptional control mechanisms[7–10,21,51,52], the present work revealed a remarkably complex RNA-based regulatory circuit governing PUL regulation in *B. thetaiotaomicron*. At the heart of this network are the RRM-1 protein RbpB and a conserved family of paralogous sRNAs that optimize fitness by orchestrating mutually exclusive expression of opposing catabolic processes (Fig. 5). Given the conservation of RRM-1 proteins[16] and R1 sequence-containing sRNAs (Supplementary Fig. 5a) across the *Bacteroidaceae*, this regulatory circuit is likely prevalent in a substantial fraction of mammalian intestinal microbiota members.

Paralogous sRNAs are ubiquitously present in the bacterial kingdom, but have mostly been studied in *Pseudomonadota* (formerly *Proteobacteria*) and *Bacillota* (*Firmicutes*)[53]. In these phyla, multicopy sRNAs are frequently involved in regulating lifestyle transitions. *E. coli* and *Salmonella enterica*, for example, encode the paralogous sRNA pair OmrA/B to reinforce the transition between motility and biofilm formation[54–57]. A pair of sibling sRNAs governs the switch from cataplerotic to anaplerotic metabolism in *Neisseria*[58–61]. The Qrr sRNA family integrates quorum-sensing signals to coordinate biofilm formation and virulence in *Vibrio* spp[62,63]. The five Cia-dependent sRNAs of *Streptococcus pneumonia* repress competence[64,65] and no less than seven sRNA paralogs (LhrC1-7) regulate virulence programs in the food-borne pathogen *Listeria monocytogenes*[66,67].

What are the functional benefits that prompt bacteria to maintain multicopy sRNAs in their genome despite the evolutionary pressure to minimize energy expenditure? Since they share a substantial degree of sequence and structure, sRNA paralogs typically regulate an overlapping set of target genes, yet may also have exclusive targets. Non-redundant regulatory functions may additionally arise from differential sRNA activation, e.g. the expression of paralogous sRNAs may be governed by different σ factors. Within the FopS sRNA cluster (excluding GibS), we did not observe strong indication of subdiversification among individual family members, except for a few SNPs in the R1' seed region of four FopS sRNAs (FopS-04, FopS-08, FopS-11, FopS-14; Fig. 3a). However, coexpression of the FopS sRNAs might have implications for dosage-dependent effects. For example, multiplexing FopS transcription across multiple loci could minimize the response time upon sensing a certain stress or nutrient cue, and rapidly rearrange cell-surface components needed to cope with hostile conditions and/or outpace metabolic competitors. Of note, the FopS

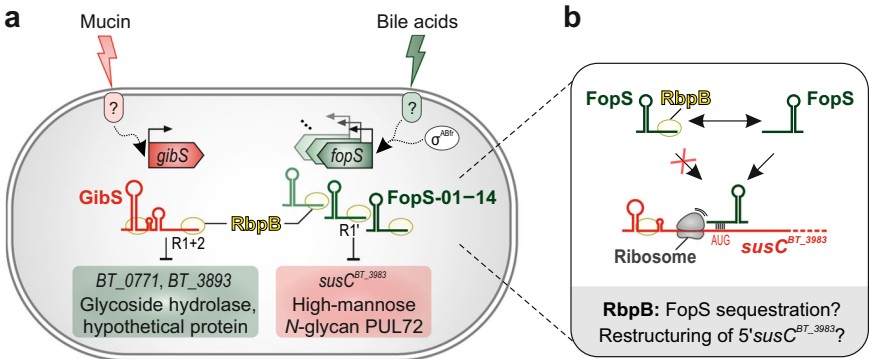

**Fig. 5 | Working model for the functional diversification between RbpB-associated sRNAs—GibS and the FopS sRNA cluster—in *B. thetaiotaomicron* metabolism control. a** GibS transcription is activated during growth on GlcNAc-containing glycans (especially mucin) through an elusive transcriptional factor recognizing a previously identified sequence motif in the *gibS* promoter ('PM2'[11]). GibS associates with RbpB (two distinct CLIP peaks) and represses the glycoside hydrolase 13 family protein BT_0771 as well as the uncharacterized protein BT_3893 through basepairing with its two distinct seeds (R1 and R2) to the translation initiation region of the corresponding mRNAs. In contrast, transcription of the 14 FopS sRNAs is driven from canonical σ$^{ABfr}$-dependent promoters and possibly further enhanced by an unknown transcriptional regulator during exposure to bile salts and potentially other membrane-assaulting stresses. The FopS sRNAs also associate with RbpB (each one CLIP peak in the 5' portion of the sRNAs). Through their extended seed sequence R1', the FopS sRNAs bind and repress translation of the mRNA for the outer membrane porin SusC$^{BT\_3983}$ of mammalian high-mannose *N*-glycan-catabolic PUL72. Given the cross-talk of the SusCD system of PUL72 with polysaccharide capsule biosynthesis[20], the FopS sRNAs may indirectly also influence the surface glycan structure of *B. thetaiotaomicron*. While the present study's focus was on PUL72, our data suggest FopS regulation to extend to numerous additional PULs (experimental support for the in-vitro interaction of FopS with seven additional PUL polycistrons [namely PUL12, -20, -44, -51, -77, -80, -88] and various further PUL genes amongst the top-scoring in-silico-predicted FopS target candidates [Supplementary Data 2]). **b** The proposed role of RbpB in post-transcriptional PUL72 regulation. In vitro, FopS-10 inhibited SusC$^{BT\_3983}$ translation and addition of recombinant RbpB was sufficient to overcome repression. In vivo, expression of *susC*$^{BT\_3983}$ was upregulated in an *rbpB* overexpression strain and decreased in the absence of the protein. This implies that RbpB counteracts FopS-mediated target repression, either by titrating the sRNA away from the target mRNA or by evoking structural changes within the translation-enhancing region of *susC*$^{BT\_3983}$.

cluster is not the only multicopy sRNA family in *Bacteroides* that is associated with RbpB. For example, the second enriched sequence motif within its ligands (Supplementary Fig. 4a) is contained in another paralogous sRNA family, consisting of six members (BTnc020, BTnc054, BTnc075, BTnc172, BTnc213, BTnc266) that are each encoded adjacent to transposases/invertases. CRISPR-based combinatorial perturbation screens that build upon existing technology[44] might in the future be leveraged to systematically dissect additive effects and functional redundancies within the FopS cluster and other paralogous sRNA families in *Bacteroides* and beyond.

The *Bacteroidota* diverged from the common line of eubacterial descent before other groups[68]. Consequently, these bacteria are fundamentally different from the other major Gram-negative phylum, the *Pseudomonadota*, with its long-standing model species *E. coli* and *S. enterica*. For example, *Bacteroides* spp. evolved unique transcription[69] and translation[70] initiation signals. Importantly, although they encode hundreds of noncoding RNAs[11–13,15], these bacteria lack the classical chaperones of pseudomonadotal sRNAs[71]. Despite recent attempts to infer *Bacteroidota* RNA chaperones from in-silico prediction and in-vitro experimentation[15,16], up to now the only global RBP that had been functionally characterized in this phylum was the transcription termination factor Rho[72].

Applying CLIP-seq, we here demonstrated that *B. thetaiotaomicron* RbpB directly associates with >170 mRNAs and ~60 sRNAs. Among the latter are the functionally characterized GibS[11], MasB[12], and BatR[44], which all act via basepairing to complementary stretches near the start codon of their respective target mRNAs. In fact, similar to the situation with GibS (Supplementary Fig. 9b), we observed ternary complexes composed of RbpB, MasB or BatR, and their respective target mRNAs in vitro (Supplementary Fig. 10c). It therefore appears plausible that RbpB may oligomerize in vivo, offering multiple binding sites for RNA. Besides, we observed sRNA-independent binding of RbpB to the 5' UTRs of selected mRNAs in vitro and according to our CLIP-seq data, RbpB preferentially binds to AU-rich sequences that are frequently found around the *Bacteroides* ribosome binding site[70]. A pathway enrichment analysis further indicated translation-associated processes to be overrepresented amongst the functional annotations of RbpB mRNA ligands. In the one tested example, the addition of recombinant RbpB did not alter the translational output of *susC*$^{BT\_3983}$ in the absence of a repressing sRNA. Structural analysis of RbpB, including the determination of its RNA-binding residues, might provide valuable insights needed to better understand the molecular mechanism(s) of this global RNA binder in the future.

In summary, the present work uncovered an RNA network controlling PUL translation in *Bacteroides*—genetic loci that are important for efficient colonization of the mammalian gut and were previously known to be regulated transcriptionally. More generally, our study serves as a model for RNA-mediated metabolism control by gut commensals and proposes RRM-1 domain-containing proteins as excellent candidates to identify additional post-transcriptional hubs in the microbiota. This knowledge could spur ongoing endeavors to exploit *Bacteroides* RNA elements as tools for microbiome engineering[73].

## Methods
### Mouse experiments
Mouse experiments were conducted in accordance with the policies of the Institutional Animal Care and Use Committee at Vanderbilt University Medical Center (protocol # M2300019-00). C57BL/6 J wild-type (cat# 000664), were obtained from Jackson Laboratory. Mice were housed in sterile cages under specific pathogen-free conditions on a 12 h light cycle, with *ad libitum* access to food and sterile water at Vanderbilt University Medical Center. Seven to nine-week-old female or male mice were randomly assigned into treatment groups before the experiment. One week before antibiotic treatment, mice were switched to a low fiber diet (TD.86489) or remained on a control diet 5010 (LabDiet 0001326) until the end of the experiment. Antibiotic cocktails (ampicillin [Sigma-Aldrich], metronidazole [Sigma-Aldrich], vancomycin [Chem Impex International], and neomycin [Sigma-Aldrich]; 5 mg of each per mouse) were administered by oral gavage daily for 5 days. After antibiotic treatment, mice were inoculated with an equal mixture of $0.5 \times 10^9$ CFU of the *B. thetaiotaomicron* wild-type strain and $0.5 \times 10^9$ CFU of the respective *rbpB* mutant for 6 days. After

euthanasia, cecal and colonic tissue was collected in sterile PBS, and the abundance of *B. thetaiotaomicron* strains was quantified by plating serial-diluted intestinal contents on selective agar.

## 16S rDNA sequencing

Total DNA was isolated from colonic contents using the Qiagen Pow-erSoil Pro Kit and the libraries were prepared by amplifying the V3-V4 hypervariable region of 16S ribosomal RNA-encoding sequences. The library was purified and sequenced on an Illumina MiSeq platform, generating 250 bp single-end reads. The downstream analysis was performed with QIIME2 version 2024.5[74]. Taxonomic profiling was performed using the SILVA taxonomy database version 132[75]. The sequencing reads generated during the current study are available at the European Bioinformatics Institute repository under accession No. PRJEB80083 (secondary accession ERP164134).

## Bacterial cultivation and genetics

Liquid cultures of *B. thetaiotaomicron* strains were prepared in complex tryptone-yeast extract-glucose (TYG) medium (20 g L$^{-1}$ tryptone, 10 g L$^{-1}$ yeast extract, 0.5% glucose, 5 mg L$^{-1}$ hemin, 1 g L$^{-1}$ cysteine, 0.0008% CaCl$_2$, 19.2 mg L$^{-1}$ MgSO$_4$·7H$_2$O, 40 mg L$^{-1}$ KH$_2$PO$_4$, 40 mg L$^{-1}$ K$_2$HPO$_4$, 80 mg L$^{-1}$ NaCl, 0.2% NaHCO$_3$) or minimal medium (1 g L$^{-1}$ L-cysteine, 5 mg L$^{-1}$ hemin, 20 mg L$^{-1}$ L-methionine, 4.17 mg L$^{-1}$ FeSO$_4$, 0.2% NaHCO$_3$, 0.9 g L$^{-1}$ KH$_2$PO$_4$, 0.02 g L$^{-1}$ MgCl$_2$·6H$_2$O, 0.026 g L$^{-1}$ CaCl$_2$·2H$_2$O, 0.001 g L$^{-1}$ CoCl$_2$·6H$_2$O, 0.01 g L$^{-1}$ MnCl$_2$·4H$_2$O, 0.5 g L$^{-1}$ NH$_4$Cl, 0.25 g L$^{-1}$ Na$_2$SO$_4$) supplemented with 0.5% of the indicated carbon sources. For propagation on plates, brain heart infusion-supplemented (BHIS) agar (52 g L$^{-1}$ BHI agar powder, 1 g L$^{-1}$ cysteine, 5 mg L$^{-1}$ hemin, 0.2% NaHCO$_3$) was used. Cultures were incubated at 37 °C in an anaerobic chamber (Coy Laboratory Products) in presence of an anoxic gas mix of 85% N$_2$, 10% CO$_2$, 5% H$_2$. *E. coli* strains were cultured aerobically in Luria-Bertani (LB) broth (10 g L$^{-1}$ tryptone, 5 gL$^{-1}$ yeast extract, 10 g L$^{-1}$ NaCl) at 37 °C with shaking or statically on LB agar plates. All bacterial strains, plasmids, and oligonucleotides used in this study are listed in Supplementary Data 3.

Defined *B. thetaiotaomicron* deletion mutants (Δ*rbpB*, Δ*fopS-08*, Δ*fopS-09*, Δ*fopS-10*, ΔΔ*fopS-09/10*) were generated using the pSIE1 plasmid system[76]. In brief, flanking regions ( ~750 bp) of the intended deletion site were assembled into the linearized pSIE1 plasmid using NEBuilder® HiFi DNA Assembly kit (NEB, E2621). Plasmids were transformed into electro-competent *E. coli* S17-1 λpir and further conjugated into *B. thetaiotaomicron*. Conjugants were selected on BHIS$^{gent/erm}$ plates. Subsequently, single colonies were inoculated in TYG medium without antibiotics and cultured overnight. The next day, cultures were serially diluted (10$^{-1}$ to 10$^{-3}$) and plated on BHIS agar plates supplemented with 100 ng mL$^{-1}$ anhydrotetracycline (aTC). Colony PCR and Sanger sequencing were performed to confirm the intended deletions. Complementation strains under control of the respective native promoter (*fopS-08*$^+$, *fopS-09*$^+$, *fopS-10*$^+$) were constructed via Gibson assembly using the pWW3452[77] plasmid (AWP-015). For over-expression of *rbpB* (strain *rbpB*$^{++}$), constructs were generated analogous to the complementation strains, except that the native promoter was replaced by a strong phage promoter. For CLIP-seq, *rbpB* was expressed in frame with a downstream 3xFLAG and 6xHis tag, stably integrated into the wild-type chromosome. Cloning gRNAs for the CRISPRi-based *fopS* knockdown was performed as described in ref. 44. Briefly, oligonucleotides containing the 20-nt spacer sequence and flanking regions on both ends 5' (5'ATCTTTGCAGTAATTTCTACTGT TGTAGAT) and 3' (5'CCGGCTTATCGGTCAGTTTCACCTGATTTA) were inserted into the *Sma*I-linearized vector (AWP-029) encoding an inactive Cas12a nuclease (dPb2Cas12a) under control of an isopropyl-β-d-1-thiogalactopyranoside (IPTG)-inducible promoter. Oligonucleotides and the linearized vector were joined in a 10 μL reaction with an oligonucleotide:vector ratio of 20:1. Assembled constructs were transformed into electrocompetent *E. coli* S17-1 λpir and subsequently

conjugated into the *B. thetaiotaomicron* double-deletion strain ΔΔ*fopS-09/10*. Constructs used for *trans*-complementation, CLIP-seq, and CRISPRi are variants of the pNBU2 integrative vector system[78], which allows their insertion into the *Bacteroides* genome at one of the two *tRNA*$^{ser}$ sites.

## UV cross-linking and immunoprecipitation (CLIP)

CLIP experiments were performed as described in ref. 79 with minor modifications. In brief, bacterial cultures were grown to mid-exponential phase (OD$_{600}$ = 2.0) in TYG. For each biological replicate experiment, 200 OD equivalents were harvested, of which 50 mL were irradiated with UV light (254 nm) at 800 mJ/cm$^2$ on a 22 cm x 22 cm plastic tray (crosslinked), while the remaining 50 mL were not (non-crosslinked control). Cells were pelleted (4000 x g, 40 min, 4 °C) and snap-frozen in liquid nitrogen. Each pellet was resuspended in 800 μL of NP-T buffer (50 mM NaH$_2$PO$_4$, 300 mM NaCl, 0.05% Tween 20, adjusted to pH 8.0). Cells were lysed mechanically in a standard Retsch apparatus (30 1/s, 10 min), using 1 mL of 0.1 mm glass beads for grinding. To clear the lysates, samples were centrifuged twice at 16,000 x g for each 15 min at 4 °C.

Cleared lysates were mixed with equal volumes of NP-T buffer containing 8 M urea and incubated for 5 min at 65 °C, with shaking at 900 rpm. Each lysate was then diluted (1:10) in pre-cooled NP-T buffer. Subsequently, 30 μL of anti-FLAG M2 magnetic beads (Sigma-Aldrich) were washed and equilibrated in 800 μL of NP-T buffer, added to the samples, and incubated for 1 h at 4 °C, rotating. Beads were collected by centrifugation at 1500 x g for 1 min at 4 °C and washed twice with each 2 mL of high-salt buffer (50 mM NaH$_2$PO$_4$, 1 M NaCl, 0.05% Tween 20, adjusted to pH 8.0) and 2 mL of NP-T buffer. Each sample was resuspended in NP-T buffer, containing 1 mM MgCl$_2$ and 25 U Benzo-nase and incubated for 10 min at 37 °C, 900 rpm, followed by a 2 min incubation on ice. After washing the beads once with 1 mL of high-salt buffer and twice with 1 mL of CIP-buffer (100 mM NaCl, 50 mM Tris-HCl [pH 7.4], 10 mM MgCl$_2$), 100 μL of CIP-mix containing 10 U of calf intestinal alkaline phosphatase (New England Biolabs) in CIP-buffer were added and beads incubated for 30 min at 37 °C, 800 rpm. Sub-sequently, one wash with 500 μL of high-salt buffer and two washes with 500 μL of 1x PNK buffer (Reaction buffer A, ThermoScientific) were performed. For labeling, a PNK mix was prepared containing 98 μL of 1x PNK buffer, 10 U of T4 polynucleotide kinase (Thermo-Scientific), and 10 μCi γ$^{32}$P-ATP. Beads were resuspended in the PNK mix and incubated for 30 min at 37 °C. Finally, 10 μL of non-radioactive ATP (1 mM) were added and samples incubated for 5 min at 37 °C, before washing the beads two times with 1 mL of NP-T buffer. For elution, beads were resuspended in 15 μL of 1x protein loading buffer with 50 mM DTT and incubated for 5 min at 95 °C. Magnetic beads were collected on a magnetic separator and the supernatant was transferred to a fresh tube. The elution step was repeated, the two supernatant fractions pooled and loaded (total volume of 30 μL) on a 15% sodium dodecyl sulfate (SDS)–polyacrylamide (PAA) gel.

Protein-RNA complexes were transferred on a Portran 0.45 μm NC membrane (Amersham). The protein ladder was labeled with a radio-active marker pen and membranes were exposed to a phosphor screen overnight. The autoradiogram was visualized, printed, and aligned with the membrane to properly excise the RNA-protein complexes from the membrane. Each membrane piece was cut into smaller pieces and transferred to a low-binding tube. To each tube, 200 μL of PK mix were added, containing 2x PK buffer (100 mM Tris-HCl [pH 7.9], 10 mM EDTA, 1% SDS), 1 mg/mL Proteinase K (Fermentas) and 10 U SUPERaseIN (Life Technologies), and incubated for 1 h at 37 °C, 800 rpm. Additionally, 100 μL of PK buffer containing 9 M urea were added and incubated for another hour at 37 °C, 800 rpm. The solution was cleared from membrane pieces and one volume of P:C:I (ROTI phenol:-chloroform:isoamyl alcohol) was added. Samples were incubated in phase-lock tubes for 5 min at 30 °C, 1000 rpm, centrifuged for 12 min at

16,000 x g, 4 °C, and the aqueous phase transferred to a fresh tube. RNA was precipitated with three volumes of 30:1 mix (ethanol:3 M NaOAc, pH 6.5) and 1 μL of GlycoBlue (ThermoScientific) overnight at -20 °C. Subsequently, RNA was pelleted, washed with 80% (vol vol⁻¹) ethanol, resuspended in 10 μL of $H_2O$ and dissolved at 65 °C for 5 min, shaking (800 rpm). Samples were stored at -20 °C until sequencing.

## CLIP-seq protocol and data analysis

The preparation of cDNA libraries and high-throughput sequencing was performed at vertis Biotechnologie AG, Freising, Germany. First, oligonucleotide adapters were ligated to the 3′ and 5′ ends, followed by a first-strand cDNA synthesis using M-MLV reverse transcriptase and the 3′ adapter as primer. The resulting cDNAs were PCR-amplified using a high-fidelity DNA polymerase with a varying number (18-24) of PCR cycles per sample. Amplified cDNA was purified using the Agencourt AMPure XP kit (Beckman Coulter Genomics). For sequencing, the samples were pooled in equimolar amounts and paired-end-sequenced on an Ilumina HiSeq system with 2x150 bp read lengths.

For analysis, raw reads were filtered and trimmed with BBDuk (paired-end mode, phred score cutoff: 20, minimal read length: 12 nt). To remove putative PCR duplicates, reads were deduplicated with FastUniq 1.1[80]. Mapping was done with the READemption 1.0.5 pipeline using segemehl 0.3.4[81]. READemption align was run in paired-end mode with an accuracy of 80%. Peak calling was performed with PEAKachu pipeline version 0.2.0[82]. First, normalization factors for peak calling were calculated with an in-house *R* script as described in ref. 28. In brief, the core positions of all libraries were isolated using an exploratory analysis of read counts summarized per position. Positions with low read counts were filtered. Normalization factors were then calculated based on the background positions, which are represented by high read counts in both crosslinked and non-crosslinked libraries. PEAKachu adaptive was run in paired-end and paired-replicates mode together with the calculated normalization factors. Maximum fragment size was set to 50 and mad multiplier to 0. Only peaks with a $log_2$-fold-change ($log_2$ FC) ≥ 2 and an adjusted *p*-value ($p_{adj}$, Benjamini-Hochberg corrected) ≤0.05 were considered significant. The annotation of those peaks and the read coverages from the sequenced CLIP-seq libraries can be retrieved through our public database, Theta-Base[12], by first clicking on the JBrowse icon and then selecting tracks from the "RbpB_CLIP" tab to the left. Users can jump to individual CLIP peaks by typing the respective peak ID into the search field at the top of the page.

As per default[79], we first considered only uniquely mapped reads in our CLIP-seq quantification and discarded multi-mapped reads. However, given the observed sequence similarity amongst the identified RbpB targets (e.g. the FopS sRNAs), we re-ran the analysis, this time also considering multi-mapped reads. While not affecting the overall results, this adaptation of the peak calling pipeline led to a further increase in the number of significantly enriched FopS sRNAs (from 10 FopS sRNAs when multi-mapped reads were discarded to all 14 FopS members when also these reads were included).

## Pathway enrichment analysis of CLIP-seq peaks

The genes containing at least one significant peak were checked for enrichment of gene sets belonging to a custom annotation (containing GO terms, KEGG pathways and modules, and functional information parsed from the literature[12]). Enrichment was computed with the enricher function of the clusterProfiler *R* package version 3.14.3[83].

## RbpB purification

Expression and purification of RbpB was performed at the recombinant protein expression facility of the Rudolf-Virchow-Center, Würzburg, Germany. In brief, constructs containing the coding sequence of RbpB together with a His-Sumo3 tag were cloned in an *E. coli* expression vector (pETM11) and transformed into *E. coli* Bl21 (DE3). For large-scale purification, cultures were grown to an $OD_{600}$ of 0.6 in

8 L of LB medium, induced with 0.5 mM IPTG, and incubated overnight at 18 °C. Cells were resuspended in 5–10 mL of lysis buffer (150 mM NaCl, 50 mM $NaH_2PO_4$, pH 7.0, 10% sucrose, 10 mM imidazole, 1 mM TCEP, 1 mM $MgCl_2$, protease inhibitor, DNase) per gram cell pellet and lysed by sonication. Lysates were cleared by centrifugation (30,000 x g, 30 min, 4 °C) and incubated with 4 mL of Ni-NTA (equilibrated to lysis buffer) for 1 h at 6-8 °C followed by immobilized metal affinity chromatography (IMAC) purification. Elution was performed seven times with each 4 mL of IMAC elution buffer (150 mM NaCl, 20 mM $NaH_2PO_4$, pH 7.0, 10% sucrose, 250 mM imidazole, 1 mM TCEP, 1 mM $MgCl_2$).

IMAC eluates were pooled and the tag was cleaved off using SenP2 Sumo-protease. Pooled eluates were dialyzed overnight against 2 L of dialysis buffer (150 mM NaCl, 20 mM $NaH_2PO_4$, pH 7.0, 10% sucrose, 1 mM TCEP, 1 mM $MgCl_2$) at 6-8 °C. Finally, the dialysate was concentrated (10 mL total volume), cleared by centrifugation (16,000 x g, 20 min, 4 °C), and loaded on a HiLoad Superdex 75 16/600 pg for size exclusion chromatography. Eluate fractions were collected (2 mL each), pooled from two runs, and analyzed by SDS-PAGE.

## In-vitro transcription and radiolabeling of RNA

For in-vitro transcription, DNA templates were amplified from genomic DNA using primer pairs carrying a T7 promoter (Supplementary Data 3). The ensuing in-vitro transcription reaction was performed using the MEGAscript T7 kit (ThermoFisher Scientific). Excess DNA was removed by DNase I digestion (1 U, 37 °C for 15 min) and the RNA product purified from a 6% (vol vol⁻¹) PAA-7M urea gel using a LowRange RNA ladder (ThermoFisher Scientific) for precise sizing. The in-vitro-transcribed RNA was eluted in RNA elution buffer (0.1 M NaOAc, 0.1% SDS, 10 mM EDTA) overnight on a thermoblock at 8 °C and 1400 rpm, subsequently precipitated using ethanol:NaOAc (30:1) mix, washed with 75% ethanol, and resuspended in 20 μL of water (65 °C for 5 min).

For radioactive labeling, 50 pmol of the in-vitro-transcribed RNA were dephosphorylated using 25 U of calf intestine alkaline phosphatase (NEB) in a 50 μL reaction volume. After 1 h incubation at 37 °C, RNA was extracted with a phenol:chloroform:isoamyl alcohol mix (P:C:I, 25:24:1) and precipitated as described above. Finally, 20 pmol of the dephosporylated RNA were 5′ end-labeled (20 μCi of ³²P-γATP) in a 20 μL reaction for 1 h at 37 °C using 1 U polynucleotide kinase (NEB). Labeled RNA was purified on a G-50 column (GE Healthcare) and extracted from a PAA gel as described above.

## Electrophoretic mobility shift assay (EMSA)

Protein-RNA and RNA-RNA EMSAs were performed in a final reaction volume of each 10 μL, containing 1x RNA structure buffer (SB; Ambion), 1 μg yeast RNA (~4 μM final concentration), and 5′ end-labeled RNA (4 nM final concentration). Reactions were either incubated with increasing concentrations of in-vitro-transcribed target mRNA segments (0; 8; 16; 32; 64; 128; 256; 512; 1,024 nM final concentration) or increasing concentrations of purified RbpB (0; 0.15; 0.3; 0.6; 1.25; 2.5; 5; 10; 20; 30; 40; 80 μM final concentration). Reactions were incubated for 1 h at 37 °C and stopped by adding 3 μL of 5x native loading dye (0.2% bromophenol blue, 0.5x TBE, 50% glycerol) and loaded on a native 6% (vol vol⁻¹) PAA gel in 0.5x TBE buffer at 4 °C and run at 300 V for 3 h. The gel was dried at 80 °C for 1 h on a Gel Dryer 583 (Bio-Rad), exposed overnight, and visualized on a phosphorimager (FLA-3000 Series, Fuji).

Three-component EMSA was carried out in a 15 μL reaction volume, containing 1x RNA structure buffer (SB; Ambion), 1 μg yeast RNA (~ 4 μM final concentration), and 5′ end-labeled RNA (4 nM final concentration). Depending on the experimental setup, reactions were either incubated with in-vitro-transcribed target mRNA segments or sRNAs (500 nM final concentration). Increasing concentrations of purified RbpB were added to the reactions (0; 0.15; 0.3; 0.6; 1.25; 2.5; 5; 10; 20 μM final concentrations).

## Rifampicin treatment to halt de-novo transcription

To assess RNA stability, single colonies of *B. thetaiotaomicron* strains AWS-001 (WT), AWS-323 (Δ*rbpB*), and AWS-218 (*rbpB++*) were inoculated in liquid TYG medium and grown for ~16 h, subsequently sub-cultured (1:100 dilution), and grown to mid-exponential phase ($OD_{600}$ = 2.0). To halt de-novo transcription, rifampicin was added (500 μg/mL final concentration). Samples were taken immediately before the addition of rifampicin (0 min) and at indicated time points after the treatment (3; 6; 9; 12; 45; 60; 90 min). Total RNA was extracted and decay assessed by northern blot analysis or qRT-PCR (see below).

## RNA extraction and removal of genomic DNA

Generally, total RNA was isolated by hot-phenol extraction from culture aliquots. To this end, 4 OD equivalents of culture were harvested, mixed with 20% vol. stop mix (95% vol $vol^{-1}$ ethanol, 5% vol $vol^{-1}$ water saturated phenol, pH >7.0), and snap-frozen in liquid nitrogen. Lysis of the bacterial cells was mediated by the addition of 600 μL of lysozyme (0.5 mg $mL^{-1}$) and 60 μL of 10% SDS, followed by an incubation for 2 min at 64 °C, before 66 μL of 3 M NaOAc (pH 5.2) were added. For extraction, phenol was added (750 μL; Roti-Aqua phenol) and samples incubated for 6 min at 64 °C, followed by the addition of 750 μL of chloroform. After centrifugation, RNA was precipitated from the aqueous phase overnight at -20 °C with twice the volume of 30:1 (ethanol:3 M NaOAc, pH 6.5). Subsequently, RNA was pelleted, washed with 75% (vol $vol^{-1}$) ethanol, and resuspended in 50 μL of $H_2O$. To remove contaminating genomic DNA, 40 μg of RNA were treated with 5 U of DNase I (Fermentas) and 0.5 μL of Superase-In RNase Inhibitor (Ambion) for 45-60 min at 37 °C in 50 μL reaction volumes. Finally, RNA was purified through phenol-chloroform extraction (ROTI phenol:chloroform:isoamyl alcohol) and resuspended in 30 μL of $H_2O$.

## Quantitative real-time PCR (qRT-PCR)

qRT-PCR was performed as described in[11]. Briefly, a reaction mix was prepared for each well of a 96-well plate, containing 10 ng of DNase I-treated RNA, 5 μL of master mix (No ROX SYBR MasterMix blue dTTP kit, Takyon), 0.1 μL of each forward and reverse primer (10 μM each), and 0.08 μL of reverse transcriptase (One-Step Kit converter, Takyon). Analysis was on a CFX96 instrument (Biorad).

## Identification of FopS paralogs and homologs

The 14 paralogous sequences that were identified by sequence homology were manually examined for the presence of annotated transcription start and termination sites on Theta-Base[12] and the existence of putative sORFs within each locus was excluded[84]. Originally, the annotated 5′ ends of FopS-01 (BTnc025) and FopS-04 (BTnc032), as well as the 3′ end of FopS-13 (BTnc188) were supported by only few RNA-seq reads[11]. We therefore manually curated the transcript boundaries to the previously identified secondary transcription start sites[11] in case of FopS-01 and FopS-04, and a 3′ end downstream of a predicted terminator hairpin in case of FopS-13 (curated sRNA boundaries marked by gray triangles in Fig. 3a). Based on these accurately defined transcript boundaries, we classified the individual FopS family members into intergenic sRNAs (*n* = 9), antisense RNAs (*n* = 4), and 3′ UTR-derived sRNAs (*n* = 1).

BLAST analysis of FopS homologs from different *Bacteroides* species was performed using the BLASTn suite and FopS-08 as a representative of the *B. thetaiotaomicron* family. Run parameters were adjusted to filter for a maximum of 250 target sequences, with a word size of 28 and a match/mismatch score of 1/-2. The resulting table was grouped by species and the Supplementary Fig. 5a was generated using the heat map function of GraphPad Prism version 10.0.0 for Windows (GraphPad Software, Boston, Massachusetts USA, www.graphpad.com). The same procedure was followed to retrieve the RRM protein-encoding homologs.

The percent identity of the 14 individual FopS paralogs of *B. thetaiotaomicron* with GibS was calculated using Clustal Omega multiple sequence alignment[85] and visualized by passing the output to MView v1.67[86].

## In-silico prediction of the FopS sRNA consensus structure

Curation of the *B. thetaiotaomicron* FopS consensus structure was done as described in detail in[15] and originally reported in[87]. Briefly, alignments of the 14 FopS sRNA sequences were generated with the WAR webserver[33] and the maximum consistency alignment and structure were downloaded in stockholm format. We then used the RALEE emacs "RNA editor mode"[88] to manually curate the alignment and optimized it with R-scape[89].

## Northern blot analysis

For northern blot analysis, 5 μg total RNA were denatured for 5 min at 95 °C, incubated on ice for 5 min, and separated on a 6% (vol $vol^{-1}$) PAA-7 M urea gel at 300 V for ~2 h. Electroblotting of the RNA onto a Hybond-N + membrane (GE Healthcare Life Sciences) was performed at 50 V, 4 °C for 1 h. Membranes were UV-crosslinked (0.12 J/$cm^2$), pre-hybridized in 15 mL of Hybri-Quick buffer (Carl Roth AG) at 42 °C for 1 h, and incubated with $^{32}$P-labeled gene-specific oligonucleotides at 42 °C. Blots were washed with decreasing concentrations (5×, 1×, 0.5×) of SSC buffer (20× SSC: 3 M NaCl, 0.3 M sodium citrate, pH 7.0), exposed as required, and visualized on a phosphorimager (FLA-3000 Series, Fuji).

## sRNA target prediction

To predict FopS targets, the IntaRNA program[41,90] was employed. The query input consisted of a 35 nt sequence comprising the complete R1′ region at a consensus of 70% (GTGTTTTTCATAGTATTAGATTTAAGG TTAACAAA) and the program was executed using its default settings against the *B. thetaiotaomicron* VPI-5482 genome.

## *B. thetaiotaomicron* growth kinetics in minimal medium containing defined carbon sources

Growth assays on different carbon sources were performed in minimal medium (1 g $L^{-1}$ L-cysteine, 5 mg $L^{-1}$ hemin, 20 mg $L^{-1}$ L-methionine, 4.17 mg $L^{-1}$ $FeSO_4$, 0.2% $NaHCO_3$, 0.9 g $L^{-1}$ $KH_2PO_4$, 0.02 g $L^{-1}$ $MgCl_2·6H_2O$, 0.026 g $L^{-1}$ $CaCl_2·2H_2O$, 0.001 g $L^{-1}$ $CoCl_2·6H_2O$, 0.01 g $L^{-1}$ $MnCl_2·4H_2O$, 0.5 g $L^{-1}$ $NH_4Cl$, 0.25 g $L^{-1}$ $Na_2SO_4$) supplemented with 0.5% of the indicated carbon sources. A single colony of the indicated strains was used to inoculate 5 mL of minimal medium-glucose and pre-cultured for 24 h. Subsequently, 1 mL of the bacterial culture was pelleted (2000 x g for 3 min) and resuspended in the same volume of minimal medium without any carbon source. This served as the inoculum (1:100) for fresh minimal medium containing either high-mannose (Man9GlcNAc2; NatGlycan LLC) or glucose. For measurement, diluted cultures were transferred to individual wells of a 96-well plate and optical density was recorded every 20 min over a time-course of 48 h.

## Inline probing

Inline probing was performed as described in ref. 11. Briefly, 0.2 pmol 5′ end-$^{32}$P-labeled RNA were incubated for 40 h at room temperature in 2x inline probing buffer (100 mM KCl, 20 mM $MgCl_2$, 50 mM Tris-HCl, pH 8.3). Reactions were stopped by the addition of 10 μL of 2x gel-loading solution (10 M urea, 1.5 mM EDTA, pH 8.0). To prepare the RNase I ladder, 0.4 pmol 5′ end-$^{32}$P-labeled RNA were denatured in 1x sequencing buffer (Ambion) at 95 °C for 1 min. RNase TI (0.1 U) was added and incubated at 37 °C for 5 min. The alkaline hydrolysis ladder was prepared by incubating 0.4 pmol 5′ end-$^{32}$P-labeled RNA in 9 μL of 1x alkaline hydrolysis buffer (Ambion) for 5 min at 95 °C. To stop the reaction, 12 μL of loading buffer II were added to both ladders and stored on ice. Samples were resolved on a 10% (vol $vol^{-1}$) PAA-7 M urea sequencing gel at 45 W for 2-3 h. Gels were dried and visualized as described above.

## In-vitro translation assay and western blotting

In-vitro translation assays were performed using a reconstituted *E. coli* protein synthesis system (PURExpress, New England Biolabs). Reactions were performed following manufacturer's instruction with a few modifications based on our prior expertise[91]. In brief, 0.5 or 1 μM in-vitro-transcribed mRNA (*5'susC^BT_3983-gfp-3xFLAG*, *gfp-3xFLAG*) were incubated in presence or absence of 25 or 50 μM in-vitro-transcribed FopS-10 sRNA for 1 min at 95 °C and chilled on ice for 5 min. RbpB (20 μM) was added and the reaction pre-incubated for 10 min at 37 °C before the PURExpress components were added. After ~4 h, reactions were stopped by adding 5 μL of 5x protein loading buffer and the whole sample volume (30 μL) was loaded on a 12% SDS-PAA gel. Proteins were transferred onto a Portran 0.2 μm NC membrane (Amersham) for 1.5 h, 350 mA at 4 °C under semi-dry conditions. To assess equal sample loading, membranes were stained with Ponceau S (Sigma-Aldrich), visualized, and then destained with 0.1 M NaOH prior to blocking in TBS-T with 10% powdered milk (1 h, room temperature). Monoclonal anti-FLAG (Sigma-Aldrich) antibody was added (1:1000 in TBS-T with 10% powdered milk) and incubated overnight at 4 °C, with shaking. Membranes were washed three times in TBS-T for each 10 min and incubated with anti-mouse IgG, HRP (ThermoFisherScientific) diluted 1:10,000 in TBS-T with 10% powdered milk for 1 h at room temperature. After rinsing, ECL detection substrate (Amersham) was added and HRP activity detected using a CCD imager (Amersham ImageQuant 800 systems).

The translational fusion of target sequences to *gfp* allowed for a fluorescence-based kinetic readout, alternatively to the above end-point determination via immunoblotting. To this end, 0.5 μM of in-vitro-transcribed FopS target mRNA variants (*5'susC^BT_3983-gfp-3xFLAG*, wild-type or point-mutated sequences) were incubated in presence or absence of 25 μM in-vitro-transcribed FopS-10 sRNA (wild-type or mutated) for 1 min at 95 °C and chilled on ice for 5 min. Components of the PURExpress Kit were added and reactions adjusted to a final volume of 12 μL. Each reaction was prepared in two technical replicates and thus split into 5 μL aliquots and distributed into individual wells of a 96-well plate (V-bottom) for fluorescence measurements. Kinetics were recorded for 16 h at 37 °C, with fluorescence measurement in 3-min intervals.

### Reporting summary

Further information on research design is available in the Nature Portfolio Reporting Summary linked to this article.

## Data availability

Sequencing data are available at NCBI Gene Expression Omnibus under the accession number GSE244816. Source data are provided with this paper.

## Code availability

Core software central to the conclusions drawn in this study are publicly available and their usage parameters described in the appropriate sections above. The CLIP-seq normalization script is described in ref. 79 and available at github[92].

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

## Acknowledgements

We would like to acknowledge the recombinant protein expression facility of the Rudolf-Virchow-Center (University of Würzburg) for expression and purification of the RbpB protein. We thank Milan Gerovac and members of the Westermann and Faber labs for fruitful discussions of this work. We are grateful to Anke Sparmann, Jörg Vogel, Chase Beisel, Lena Amend, and Thomas Guest for critical reading and constructive feedback on previous drafts of this manuscript. This work was funded by the German Research Foundation (DFG; Individual Research Grant We6689/1-1 to A.J.W.). Research in the Westermann laboratory is supported by the European Research Council (ERC Starting Grant #101040214).

## Author contributions

Conceptualization, A.-S.R., D.R., and A.J.W.; methodology, A.-S.R., D.R., L.S., V.L.-S., G.P., L.B., F.F., W.Z., and A.J.W.; investigation, A.-S.R., D.R., L.S., V.L.-S., G.P., S.R., and M.L.; writing – original draft, A.-S.R. and A.J.W.; writing – review and editing, A.-S.R., D.R., G.P., L.B., F.F., W.Z., and A.J.W.; funding acquisition, A.J.W.; resources, L.B., F.F., W.Z., and A.J.W.; supervision, A.J.W.

## Funding

## Competing interests

The authors declare no competing interests.
