## [Peer Review file · Nature Communications]

The global RNA-binding protein RbpB is a regulator of polysaccharide utilization in *Bacteroides thetaiotaomicron*

Corresponding Author: Professor Alexander Westermann

Version 0:

Reviewer comments:

Reviewer #1

(Remarks to the Author)

This manuscript by Rüttiger et al. provides a comprehensive and well-structured study of the proposed role of RbpB as a global RNA binder that directly interacts with numerous cellular transcripts, including multicopy sRNA families (FopS and GibS) in prominent gut microbes. This complex post-transcriptional regulation allows *Bacteroides* to switch off certain PULs in the absence of specific substrate sources to optimize fitness in the mammalian gut environment to adapt to different dietary conditions. The study's design is robust and combines a variety of experimental techniques with detailed molecular analyses to assess the role of RbpB. The manuscript also provides an in-depth analysis of the interaction between RbpB and RNA, identifying specific binding sites and motifs and the identification of conserved multicopy sRNA families associated with RbpB. The proposed mechanism of the global RNA-binding protein RbpB promoting mucus-foraging in *Bacteroides* is significant and contributes to the understanding of how prominent gut microbes prevent ongoing translation of pre-existing PUL mRNAs once the inducing stimuli disappear, which could be relevant for microbiome research generally and potential therapeutic applications in particular. While the manuscript is detailed and thorough, there are some aspects where improvements could enhance the clarity of the study for the reader, as follows.

*Lines 75-76. "While the 14 FopS members of *B. thetaiotaomicron* exhibit partial sequence similarity to the previously characterized GibS sRNA". Please indicate how similar those are, e.g., by indicating ID/similarity seq percentages.

*Lines 98-99. "the Δ rbpB mutant displayed a significant fitness disadvantage compared to the wild-type when switched to a low fiber diet that forced bacteria to feed on host-derived glycans".

Also, Line 107. "These results suggest RbpB to promote the mucus-foraging lifestyle of *B. thetaiotaomicron*". Considering this, it would be interesting to analyze expression levels of RbpB in *Bacteroides* grown in mucins as unique carbon source.

*Lines 242-244. "For further characterization of FopS-mediated target control, we henceforth focused on PUL72, which is functionally and structurally characterized and for which the glycan substrate—namely high mannose N-glycan—is known". Considering the authors chose PUL72 because the substrates are known, they missed an opportunity to show transcriptional/post-transcriptional analysis in the presence of high-mannose substrates, as well as expression of FopS/GibS in the presence of a high mannose diet as the sole carbon source?

*The authors described the regulation of SusC BT3983 and 3984 by FopS. What happens with the rest of the PUL (BT3983-3984 is described as an operon, but those genes are part of a PUL from 3983-3994)? Do FopS sRNAs repress PUL72 by base-pairing to the cognate susCD operon but not other components of the PUL?

*Lines 343-344. The authors mention that the expression of paralogous sRNAs may be governed by different σ factors. What is the possible interplay between σ and anti- σ factors regulating PUL72 and FopS and RbpB?

*Lines 174-175. "This is supported by size-exclusion chromatography analysis, in which RbpB behaved as a homodimer in vitro (Suppl. Fig. S6b)". The SEC analysis in Figure S6 is insufficient. The authors should re-run both the monomer and dimer fractions in order to see if they re-separate into mixtures or persist as monomers or dimers. It would be helpful to see an SDS-PAGE of the SEC fractions. No calibration curve of SEC is shown. This is related to Fig. 2a, in which there is more than one band in the fraction corresponding to RbpB-FLAG, which may be indicative of different oligomeric states.

*Line 843, "Size-exclusion chromatography. Shown is the elution profile of RbpB, with two separate elution peaks at 68 and 80 mL, indicative of RbpB monomers and dimers". Graphic is in minutes not mL. Only SUMO-RbpB fusion is shown, not RbpB alone after digestion with SUMO protease. Which fraction did they use for the experiments, monomer or dimer?

*Figure 4b. Other FopS target candidates' description. There are no comments about the post-transcriptional regulation of structurally and functionally characterized alpha-fucosidase (BT2970), putative chitinase (BT1632) and beta-galactosidase BT4684. Why does FopS modulate GHs but not SusC of these PULS?

*Figure 3d shows repression of FopS in starvation conditions, but in the northern blot (Fig. 3e) FopS exhibits a stronger signal in the stationary phase than in other growth phases. How do the authors explain this?

-The authors could suggest which is the possible sensor of bile acids and mucins indicated in *Bacteroides* cells in Figure 5a.

-The authors could discuss the relevance of structural analysis of this global regulator, dimerization/oligomerization interface, mutagenesis etc. that might be important to determining molecular mechanism of RbpB.

Reviewer #2

(Remarks to the Author)

This study analyzes an RNA-binding protein of *Bacteroides thetaiotaomicron*, elucidation of its targets, and some of the PUL that it influences. The manuscript does not have a central focus and therefore, it is difficult to extract the main findings and the relevance of the study. Some of the conclusions are strongly supported by experimentation, but others are weaker and need to be toned down or more data needs to be provided to support them. In general, the manuscript would benefit from a restructuring, eliminating the claims that are not strongly supported or that speak beyond the data, with the addition of new data to better support some findings. Many of the figures would be easier to interpret with additional information in the text and figure legends. There are numerous suggestions below to improve the manuscript.

1) The title should change the word "Bacteroides" to "*Bacteroides thetaiotaomicron*" as this study shows rbpB is not conserved in *Bacteroides*. Also, the data have not shown that RbpB is a central regulator of PS utilization, but rather utilization of a few PS.

2) Lines 30-31, the data do not support that RbpB interacts specifically with several hundred cellular transcripts.

3) Line 34, This line must be removed or clarified to state that these findings expand on previous work to show the global....

4) Line 39 –Members of the order Bacteroidales are anaerobes, but here are many facultative anaerobes, that prefer to grow aerobically in the phylum Bacteroidota. I suggest limiting this sentence to the order Bacteroidales

5) Line 41 – there are numerous factors that contribute to the success of these bacteria in the human gut. This should be rewritten such as "An important contributing factor....."

6) Line 44-45 – some PS are transported across the outer membrane intact. Maybe include the word "typically" in this statement when discussing a typical PUL.

7) Line 49 – modify to - PUL "often" include....

8) Line 65 – please replace the word "their" with what you are discussing. Is it "sRNA-based post-transcriptional regulatory networks"?

9) Line 89 – remove the word "proposed". In the cited study, RbpB was shown to bind to single-stranded RNA.

10) There is no mention in the methods or anywhere how the rbpB deletion mutant was created. Please write a methods section with an explanation of the construction of all the genetic constructs that were created in this study.

Details about the mouse experiment shown in Figure 1 are lacking

11) It is stated in the methods and figure that the WT and rbpB mutant were differentiated from the mice by selective plating. How were these two strains differentiated by plating? Plating on what? Were different antibiotic genes inserted into each strain? Details must be provided. In addition, different antibiotic resistance genes confer different fitness benefits/detriments, so gene swaps must be performed.

12) Was a total of 10⁹ CFU bacteria really gavaged into mice? More details about this procedure need to be supplied. In what volume? That is a very high inoculum.

13) The mice obviously could not be maintained on the antibiotic treatment after introduction of the *B. thetaiotaomicron* due to the inclusion of metronidazole, what was the composition of the community when the samples were collected? This is important as the low fiber and control diet may have allowed differential expansion of other members of the microbiota that influenced colonization of the Bt strains. Please provide a 16S rRNA gene analysis of the fecal samples at the end of the

experiment.

14) How did you confirm that all bacteria that grew on the “selective” plates were the particular strain of *B. theta* and not a strain of the microbiota that rebounded?

15) Lines 112-113 – remove the phrase “- as opposed to a test tube-“ as you previously said the study was performed in vitro and this is unnecessary. How was the epitope tagged RbpB created? The first section of the Experimental methods section is titled “Bacterial cultivation and genetics” but there is no mention of how any of the genetic constructs were made. The information contained in Table S2 is not enough. How was the DNA amplified, were the plasmids sequenced to ensure they did not introduce a mutation in surrounding genes.

16) There is a Table S2, but no Table S1.

17) Line 114 – state what the epitope is. State it is a FLAG-tagged version of RbpB. List here if it is N-terminal or C-terminal tag.

18) Was the flag-tagged RbpB that was introduced into the chromosome done so in the delta *rbpB* mutant? If not, the endogenous RbpB should compete for binding. Or was the tag added to the end of the native gene? Did you show that the flag-tagged RbpB was not impaired for function? You need to show that it can complement the deletion mutant for one of the various functions.

19) Line 422 – An OD600 of 2.0 is not mid-log phase. What is the basis of this statement?

20) What does “200 OD equivalents per condition” mean?

21) Line 424 – what was the volume of bacteria that were irradiated in this 22cm x 22cm plate?

22) The data presented in Fig S1C are not easy to interpret. It seems there should be a way to show the interacting RNAs that are common between the two replicates.

23) Figure S3 is confusing based on what is written in the legend and text. How does figure S3B show that the “cellular half-lives of the top-enriched mRNA ligands” are reduced when *rbpB* is deleted or overexpressed?

24) Line 155 – It is stated that “A BLAST search revealed this multicopy sRNA family to be highly conserved within the Bacteroidota (Suppl. Fig. S5a)” yet in this figure you are only showing strains of one family, Bacteroidaceae. You are not even showing other families of the order Bacteroidales. Even within the Bacteroides, RbpB is not conserved as it is absent from *B. ovatus* and *B. fragilis*. The figure legend must also be changed.

25) Line 177 – why does the inverting the 55 nt-long RbpB binding site reduce formation of the higher order complex but not the not the complex itself? Also, the formation of the higher order complex was not abrogated but rather reduced.

26) It is stated that “EMSAs also validated that RbpB binds to GibS, again forming higher order complexes (Suppl. Fig. S4b, 180 Suppl. Fig. S6e).” I don't see how Fig S4b shows this finding. In figure S6C, there is less radiolabel of the FopS-10 mutant RNA compared to WT, therefore, as the trend seems the same as in the WT, the conclusions that the mutation “was sufficient to abrogate formation of the higher order RNP complex” is not supported.

27) Lines 211-214 – the data of in support of this in Fig S7C are weak and there is no quantification.

28) Line 316 – Bacteroides is not the predominant genus in many individuals. This should be change to “one of the predominant genera...”

29) Line 323 – Based on the data shown in Fig S5A, you should not state that RbpB is conserved, or limit the statement to *B. thetaiotaomicron*.

30) Line 380 – maybe alter this statement to remove the words “urgently needed”.

31) Line 382 – not sure the study uncovered a large RNA network controlling PUL translation in Bacteroides”.

Reviewer #3

(Remarks to the Author)

In this manuscript, Rüttiger et al studied the regulatory role of the RNA binding protein, RbpB, in the human commensal bacterium *Bacteroides thetaiotaomicron*. Previous work indicated that RbpB binds to RNA in *B. thetaiotaomicron* (PMID: 34251866), however, the identity of the RNA ligands and potential regulatory functions remained unknown. Here, the authors show that RbpB binds to hundreds of transcripts in *B. thetaiotaomicron*, including a group of paralogous small RNAs (sRNAs), called FopS (1-14). Further investigations into the regulatory roles of the FopS sRNAs revealed a conserved sequence motif required for recognition by RbpB and for base-pairing. Surprisingly, FopS and RbpB have opposing functions in regulation of the *susCD* operon involved in carbohydrate utilization. The manuscript is interesting, well-written,

and the rationale of the experiments is easy to follow. I have only few comments and suggestions that I listed below.

Major points:

- Lines 128-129 and Fig. 2d: please provide additional information why the shorter peaks in the CLIP-Seq analysis are associated with mRNAs, whereas longer peaks are linked to sRNAs (is that also true for 5' and 3' UTRs?). Are there several RNA bindings sites on the protein that could indicate the simultaneous interaction of two RNA ligands?
- Lines 189-203, Fig 3: please clarify if all FopS sRNAs have their own promoters or if they are co-expressed with an upstream gene (like FopS-14).
- Figs. 4a and c: the selection of fopS deletion mutants that were created seems somewhat random as it remains unclear if FopS 09, 10, 13 are more likely to regulate these target mRNAs, when compared to the other FopS sRNAs. Therefore, it would be useful to create a mutant strain lacking all fopS sRNA genes. Alternatively, the authors could consider to over-express a synthetic antisense RNA that is complementary to all FopS sRNAs and test target regulation.
- Figs. 4g and f: I think it would be useful to also include point mutations in the base-pairing site of FopS and susC as additional controls in these experiments.

Minor points:

- Line 125: please provide additional information of the 14 intergenic peaks identified in the CLIP-Seq analysis. Are these additional, previously unidentified, sRNAs?
- Line 145: the 41 nt-long motif is somewhat confusing as the binding peaks for mRNAs (the majority of interaction partners) are only ~30nt long.
- Line 180: I think it would be useful to provide additional information on GibS at this point (or in the introduction). Otherwise, this paragraph is difficult to follow.
- Paralogous sRNAs are often regulated by a universal sponge sRNA. Is there evidence for such a sponge sRNA in *B. thetaiotaomicron*?

Version 1:

Reviewer comments:

Reviewer #1

(Remarks to the Author)

The authors have done an exceptional job in responding to reviewers' comments, addressing all major concerns, adding new experimental results, and enhancing the context of their findings.

Reviewer #2

(Remarks to the Author)

The authors addressed every one of my concerns with text modifications or the inclusion of new experimentation. I am satisfied with the revisions.

Reviewer #3

(Remarks to the Author)

The authors convincingly addressed my previous comments. I think the manuscript is now ready for publication.

Point-to-point response to reviewer comments

We would like to thank the reviewers for the time and effort they dedicated to the review of our manuscript! Their comments were well-taken and proved very helpful in our endeavor to improve the present study.

As part of this revision, we performed various additional experiments recommended by the reviewers to corroborate our conclusions, as will be outlined in detail in our responses to the individual reviewer comments below. This includes additional mouse experiments to support the specificity of the *rbpB* deletion phenotype through trans-complementation and 16S profiling to put the phenotype in context of the resident microbiota's composition. Besides, we constructed and analyzed the effect of point mutations and the respective compensatory mutations for the interaction of FopS-10 with *BT_3983* mRNA. We performed experiments to validate additional FopS target candidates, generated a multi-*fopS* knockdown mutant and identified an associated phenotype during growth on high-mannose *N*-glycans. Collectively, these new data backed up our initial claims.

Reviewer #1 (Remarks to the Author):

This manuscript by Rüttiger et al. provides a comprehensive and well-structured study of the proposed role of RbpB as a global RNA binder that directly interacts with numerous cellular transcripts, including multicopy sRNA families (FopS and GibS) in prominent gut microbes. This complex post-transcriptional regulation allows *Bacteroides* to switch off certain PULs in the absence of specific substrate sources to optimize fitness in the mammalian gut environment to adapt to different dietary conditions. The study's design is robust and combines a variety of experimental techniques with detailed molecular analyses to assess the role of RbpB. The manuscript also provides an in-depth analysis of the interaction between RbpB and RNA, identifying specific binding sites and motifs and the identification of conserved multicopy sRNA families associated with RbpB. The proposed mechanism of the global RNA-binding protein RbpB promoting mucus-foraging in *Bacteroides* is significant and contributes to the understanding of how prominent gut microbes prevent ongoing translation of pre-existing PUL mRNAs once the inducing stimuli disappear, which could be relevant for microbiome research generally and potential therapeutic applications in particular. While the manuscript is detailed and thorough, there are some aspects where improvements could enhance the clarity of the study for the reader, as follows.

Reply: Thank you very much for the assessment of our work and your constructive comments.

*Lines 75-76. "While the 14 FopS members of *B. thetaiotaomicron* exhibit partial sequence similarity to the previously characterized GibS sRNA". Please indicate how similar those are, e.g., by indicating ID/similarity seq percentages.

Reply: The percent identity of the 14 individual FopS paralogs with GibS varies between 28.4% up to 43.9%. We now include this information in the legend to Fig. 3a.

*Lines 98-99. "the Δ rbpB mutant displayed a significant fitness disadvantage compared to the wild-type when switched to a low fiber diet that forced bacteria to feed on host-derived glycans". Also, Line 107. "These results suggest RbpB to promote the mucus-foraging lifestyle of *B.*

thetaiotaomicron”. Considering this, it would be interesting to analyze expression levels of RbpB in *Bacteroides* grown in mucins as unique carbon source.

Reply: The mRNA levels of *rbpB* were quantified in a recent RNA-seq study, in which we profiled conditional gene expression of *B. thetaiotaomicron* grown in presence of different carbon sources (PMID: 38528147). A screenshot showing the read coverages over the *rbpB* genomic locus across a selection of different carbon sources is depicted below (Rebuttal Fig. 1a). In porcine mucin-containing minimal medium (“Muc”), *rbpB* is decently—albeit not maximally—expressed. In contrast, in the complete absence of a carbon source (“Starv” in the figure below), *rbpB* transcription is shut off.

As mRNA levels may not necessarily reflect translation of the corresponding protein product, we also reanalyzed published Ribo-seq data derived from mouse fecal pellets (PMID: 31402174), which provides evidence for RbpB translation in vivo (Rebuttal Fig. 1b).

Rebuttal Figure 1: a, Screenshot of read coverages across the genomic *rbpB* locus upon growth on different carbon sources. Glu, glucose; Malt, maltose; Muc, porcine mucins; NAG, N-acetylglucosamine; Starv, starvation (i.e. deprivation of a carbon source for 2 h). The *rbpB* coding sequence is indicated as a thick red horizontal line at the top. Data are derived from PMID: 38528147. **b**, Read coverages at the genomic *rbpB* locus derived from Ribo-seq datasets generated by Amy Bhatt’s group (PMID: 31402174). Raw data were downloaded from BioProject ID PRJNA540869 and aligned to the VPI-5482 genome. The track depicts two replicates that are overlapped and displayed alongside the transcription start site (‘TSS’) and terminator (‘term’) of the *rbpB* mRNA.

*Lines 242-244. “For further characterization of FopS-mediated target control, we henceforth focused on PUL72, which is functionally and structurally characterized and for which the glycan

substrate—namely high mannose N-glycan—is known”. Considering the authors chose PUL72 because the substrates are known, they missed an opportunity to show transcriptional/post-transcriptional analysis in the presence of high-mannose substrates, as well as expression of FopS/GibS in the presence of a high mannose diet as the sole carbon source?

Reply: Thank you for this comment! In fact, we had ourselves already considered conducting experiments in high-mannose *N*-glycans as sole carbon source. However, due to the complex structure of this sugar, commercial synthesis is very laborious, hence costly, and only few companies sell purified high-mannose *N*-glycans. We had therefore initially decided against conducting experiments that involve this sugar. However, triggered by the reviewer’s comment, we now purchased 10 mg of the sugar, which was sufficient for a small-scale experiment. Specifically, we monitored growth of *B. thetaiotaomicron* wild-type and a multi-*fopS* depletion mutant (see below; response to comment by Reviewer #3; pages 13-14) in minimal medium containing this substrate as sole carbon source and found the mutant to show an enhanced growth. Concomitant expression profiling suggested this growth advantage to be due to the de-repression of PUL72 upon inhibition of FopS. We include these new data in Fig. 4d and Extended Data Fig. 8d-f (northern blot of FopS/GibS expression, growth curves, qRT-PCR measurement of FopS/GibS target levels) and describe our findings in lines 271-280 of the revised manuscript.

*The authors described the regulation of *SusC* BT3983 and 3984 by FopS. What happens with the rest of the PUL (BT3983-3984 is described as an operon, but those genes are part of a PUL from 3983-3994)? Do FopS sRNAs repress PUL72 by base-pairing to the cognate *susCD* operon but not other components of the PUL?

Reply: This is a very interesting question. To address it, we have now measured the mRNA levels of all PUL72 genes individually by qRT-PCR. As shown in new Extended Data Fig. 7b, the expression of *BT_3986*, *-87*, and *-88* was also affected by FopS deletion, albeit less prominently than that of the corresponding *susCD* pair (*BT_3983/84*). Based on our previous transcriptome annotation (PMID: 38528147), there is one transcriptional start site from which two distinct operons can be transcribed, namely a short (*BT_3983/84*) and an extended primary transcript (*BT_3983-88*). Hence, FopS binding to the 5’ region of *BT_3983* (the first gene in either operon) seems to affect the abundance of both these operons. Supporting this assumption, expression of the remaining PUL72 genes—which are not co-transcribed with *BT_3983*—were not affected by *fopS* deletion.

Interestingly, we observed that overexpression of RbpB did affect the levels of some other PUL72 transcripts, namely *BT_3992-91* and *BT_3987*. We speculate that the latter might be due to direct binding of the protein to those transcripts as reflected in the RbpB CLIP-seq data (see positions of the grey vertical lines in the PUL72 locus representation at the top of Extended Data Fig. 7b).

*Lines 343-344. The authors mention that the expression of paralogous sRNAs may be governed by different σ factors. What is the possible interplay between σ and anti- σ factors regulating PUL72 and FopS and RbpR?

Reply: We deliberately remain vague in this regard, as little information can be inferred from the literature that would justify educated guesses. We currently believe that FopS transcription is at least partially mediated by the housekeeping sigma-factor of *Bacteroides*, σ^{ABfr} , since the corresponding recognition motif is found in the promoter region of most of the FopS family members (see Fig. 3a). Transcriptional control of PUL72 is governed by the ECF sigma/anti-sigma factor pair *BT_3992/93*. However, the sequence motif recognized by this latter sigma factor is elusive and hence, we do not

currently know if and how transcriptional activation of PUL72 would cross-talk with either the FopS family or RbpB.

*Lines 174-175. “This is supported by size-exclusion chromatography analysis, in which RbpB behaved as a homodimer in vitro (Suppl. Fig. S6b)”. The SEC analysis in Figure S6 is insufficient. The authors should re-run both the monomer and dimer fractions in order to see if they re-separate into mixtures or persist as monomers or dimers. It would be helpful to see an SDS-PAGE of the SEC fractions. No calibration curve of SEC is shown. This is related to Fig. 2a, in which there is more than one band in the fraction corresponding to RbpB-FLAG, which may be indicative of different oligomeric states.

*Line 843, “Size-exclusion chromatography. Shown is the elution profile of RbpB, with two separate elution peaks at 68 and 80 mL, indicative of RbpB monomers and dimers”. Graphic is in minutes not mL. Only SUMO-RbpB fusion is shown, not RbpB alone after digestion with SUMO protease. Which fraction did they use for the experiments, monomer or dimer?

Reply (to both comments): We repeated the SEC analysis (see Rebuttal Figure 2, below), which shows that in vitro, RbpB behaves as a monomer; at least in the absence of RNA ligands. We used monomeric RbpB throughout this study. In parallel, as judged from preliminary split-luciferase assays currently conducted in our lab, RbpB can homodimerize in vivo. RNP formation and structural analysis of RbpB will be the specific focus of a follow-up study recently started in our lab in collaboration with a structural group. For the present manuscript, we toned down any speculations about RbpB oligomerization and updated the corresponding Extended Data Figure as per the reviewer’s request.

Rebuttal Figure 2: SEC-MALS analysis of RbpB. Measurements were performed using an Agilent 1260 Infinity II coupled to a Superdex 75 Increase 10/300 GL, a miniDAWN multi-angle light scattering detector and an Optilab refractometer (Wyatt Technology). The system was calibrated to the buffer using BSA (2 mg/mL).

*Figure 4b. Other FopS target candidates’ description. There are no comments about the post-transcriptional regulation of structurally and functionally characterized alpha-fucosidase (BT2970), putative chitinase (BT1632) and beta-galactosidase BT4684. Why does FopS modulate GHs but not SusC of these PULS?

Reply: In the revised manuscript, we now explicitly name these glycoside hydrolases and cite existing literature on the characterization of the fucosidase. In addition, we performed additional EMSA experiments that supported the predicted interaction of FopS-10 with eight additional target

candidates (including the mRNAs for those three GHs as well as of five additional SusC homologs from various PULs). These new data is now presented in Extended Data Fig. 7e.

As to the question of why in these cases a GH—rather than the cognate SusC homologue—seems to be targeted by FopS: close inspection of the cognate PULs (namely, PUL20 and PUL44) revealed an atypical genomic organization. While PULs are typically transcribed from operons with *susCD* being the first mRNAs in the respective polycistrons, according to PULDB (PMID: 29088389) and our own previous annotations of transcription start and termination sites (PMID: 38528147), the genes for the predicted FopS target GHs are the 5'-most genes in PUL20 and -44 (adjacent to the separately transcribed genes encoding the cognate transcriptional regulatory system, e.g. ECF- σ /anti- σ factor pairs). We thus speculate that FopS sRNAs may generally bind to the 5' region of the first mRNA within their target PUL polycistrons. We have added a respective comment to our revised manuscript (see lines 298-306).

*Figure 3d shows repression of FopS in starvation conditions, but in the northern blot (Fig. 3e) FopS exhibits a stronger signal in the stationary phase than in other growth phases. How do the authors explain this?

Reply: Figure 3d shows reanalyzed RNA-seq data collected in one of our previous studies (PMID: 38528147). The heat map reports the fold-changes in expression relative to the growth in minimal medium containing glucose as the sole carbon source. The “Starvation” condition in this experiment relates to the initial growth of *B. theta* in defined minimal medium supplemented with glucose to late exponential phase (17 h after sub-culturing at a 1:100 dilution), followed by a medium exchange and an additional incubation of the bacteria for 2 hours in carbon source-deprived minimal medium. In contrast, the “Stat” sample loaded on the northern blot in Figure 3e stems from *B. theta* grown in rich TYG medium for 10 hours, reaching stationary phase ($OD_{600} = 4$). Hence, the “Starvation” and “Stat” conditions are quite different from one another and the way that the data are plotted in Fig. 3d (fold-change in expression rel. to the glucose condition) vs. Fig. 3e (absolute expression levels) further impedes a direct cross-comparison.

As outlined in our response to your comment above (page 3), we assume that FopS transcription is mediated by the housekeeping sigma-factor of *Bacteroides* (σ^{ABfr}). In stationary phase—reminiscent of the situation in *Enterobacteriaceae*, where RpoS replaces RpoN—*B. theta* RNA polymerase may substitute its σ^{ABfr} subunit with an elusive stationary phase-specific sigma-factor, which would abrogate FopS de-novo transcription. However, given the exceptionally long half-life of FopS (see Extended Data Fig. 3c), steady-state levels of these sRNAs will remain high for quite some time (explaining the pattern seen on the northern; Fig. 3e).

For the starvation condition (Fig. 3d), bacteria were initially grown in glucose-containing minimal medium, where FopS sRNAs are decently expressed (see e.g. Extended Data Fig. 8e). When then starved for 2 h in the absence of any carbon source, *B. theta* is known to induce its stringent response (PMID: 30008292; 38528147) to reallocate resources away from growth and favor persistence. We hypothesize that under those stress conditions, FopS (and other sRNAs) may be actively degraded to recycle their ribonucleotides for essential cellular processes, providing a possible explanation for why the levels of FopS are so low in this condition.

-The authors could suggest which is the possible sensor of bile acids and mucins indicated in *Bacteroides* cells in Figure 5a.

Reply: With respect to bile sensors, a recent paper from Jean-Marc Ghigo's group (one of the leading labs when it comes to the influence of bile on *Bacteroides* spp.) concludes: "We do not yet know how bile is sensed by the [*B. thetaiotaomicron*] cell..." (PMID: 38534200). In case of mucins, several PUL-associated sensory/regulatory proteins come to mind, e.g. the ECF sigma/anti-sigma factor pairs of PUL14 (BT_1052/53), -27 (BT_2169/71), -72 (BT_3992/93), and -78 (BT_4248/50), as well as the CRP-like transcription factor of PUL80 (BT_4300), which are all PULs specifically induced when *B. thetaiotaomicron* feeds on mucin glycans (PMID: 18996345; 26392213; 38528147). However, in absence of any evidence for the involvement of those proteins in FopS expression control, we refrain from making adaptations to our working model in Fig. 5a.

-The authors could discuss the relevance of structural analysis of this global regulator, dimerization/oligomerization interface, mutagenesis etc. that might be important to determining molecular mechanism of RbpB.

Reply: Thanks for pointing this out! We have added a corresponding sentence to the Discussion section (lines 422-424).

Reviewer #2 (Remarks to the Author):

This study analyzes an RNA-binding protein of *Bacteroides thetaiotaomicron*, elucidation of its targets, and some of the PUL that it influences. The manuscript does not have a central focus and therefore, it is difficult to extract the main findings and the relevance of the study. Some of the conclusions are strongly supported by experimentation, but others are weaker and need to be toned down or more data needs to be provided to support them. In general, the manuscript would benefit from a restructuring, eliminating the claims that are not strongly supported or that speak beyond the data, with the addition of new data to better support some findings. Many of the figures would be easier to interpret with additional information in the text and figure legends. There are numerous suggestions below to improve the manuscript.

Reply: Thank you for this detailed feedback and for your constructive criticism that has helped us to improve this study. We refrained from completely restructuring the manuscript, as it was well received by the other two reviewers (Reviewer #1 explicitly mentions that the study is well-structured [see above, page 1] and Reviewer 3 states that "The manuscript is interesting, well-written, and the rationale of the experiments is easy to follow" [page 12]). However, we now provide additional experimentation to support our main conclusions, more detailed information that should contribute to a better understanding of the presented data, and have amended the text according to your suggestions.

1) The title should change the word "Bacteroides" to "Bacteroides thetaiotaomicron" as this study shows rbpB is not conserved in Bacteroides. Also, the data have not shown that RbpB is a central regulator of PS utilization, but rather utilization of a few PS.

Reply: We changed the title accordingly.

2) Lines 30-31, the data do not support that RbpB interacts specifically with several hundred cellular transcripts.

Reply: The abstract was changed accordingly.

3) Line 34, This line must be removed or clarified to state that these findings expand on previous work to show the global....

Reply: This was rephrased as per the reviewer's request.

4) Line 39 –Members of the order Bacteroidales are anaerobes, but here are many facultative anaerobes, that prefer to grow aerobically in the phylum Bacteroidota. I suggest limiting this sentence to the order Bacteroidales

Reply: Thank you for this suggestion! We changed the text accordingly.

5) Line 41 – there are numerous factors that contribute to the success of these bacteria in the human gut. This should be rewritten such as “An important contributing factor.....”

Reply: This sentence was rephrased according to your suggestion.

6) Line 44-45 – some PS are transported across the outer membrane intact. Maybe include the word “typically” in this statement when discussing a typical PUL.

Reply: The word “typically” was added.

7) Line 49 – modify to - PUL “often” include....

Reply: This was changed accordingly.

8) Line 65 – please replace the word “their” with what you are discussing. Is it “sRNA-based post-transcriptional regulatory networks”?

Reply: Yes, we meant to refer to post-transcriptional regulatory circuits. This is now explicitly mentioned for improved clarity.

9) Line 89 – remove the word “proposed”. In the cited study, RbpB was shown to bind to single-stranded RNA.

Reply: This was changed accordingly.

10) There is no mention in the methods or anywhere how the rbpB deletion mutant was created. Please write a methods section with an explanation of the construction of all the genetic constructs that were created in this study.

Reply: Thank you for pointing this out. We apologize for not having included this important information in the previously submitted manuscript version. The respective information is now contained in the Methods section of the revised manuscript (see lines 449-474).

Details about the mouse experiment shown in Figure 1 are lacking

11) It is stated in the methods and figure that the WT and rbpB mutant were differentiated from the mice by selective plating. How were these two strains differentiated by plating? Plating on what? Were different antibiotic genes inserted into each strain? Details must be provided. In addition, different antibiotic resistance genes confer different fitness benefits/detriments, so gene swaps must be performed.

Reply: We thank the reviewer for this comment. The wild-type strain and the isogenic *rbpB* mutant were marked with integrated plasmids pNBU2-*bla-ermG* and pNBU2-*bla-tetQ*, respectively. These plasmids confer resistance against erythromycin and tetracycline, respectively. Of note, the *in vivo* experiments were performed in the absence of either of the antibiotics. While under some specific conditions, antibiotic-resistant cassettes could confer fitness benefits/detriments, multiple independent investigations showed that the two plasmids we used did not appreciably impact strain fitness *in vitro* and *in vivo* (PMID: 34662212, 32937127, 36757366). This is further supported by our additional experiments performed during revision, where we observed that the fitness defect of the Δ *rbpB* mutant can be completely complemented by inserting the *rbpB* gene back using the same pNBU2-based plasmid (see Fig. 1). As such, this data suggests that the plasmids we used did not skew the fitness of the strains we tested.

12) Was a total of 10^9 CFU bacteria really gavaged into mice? More details about this procedure need to be supplied. In what volume? That is a very high inoculum.

Reply: The reviewer is correct in that we inoculated a total of 100 μ l, 10^9 CFU into each mouse. This is a widely adopted practice in studies of *Bacteroides* fitness *in vivo* (PMID: 32075741, 36757366, 28943327, 38110423). Importantly, we utilized a competitive fitness experimental setup, in which each mouse received an equal mixture of two strains. As such, the two strains experience the same nutritional environment in each mouse, and the quantity of the inoculum will not likely have a significant impact on the relative fitness of each strain.

13) The mice obviously could not be maintained on the antibiotic treatment after introduction of the *B. theta* strain due to the inclusion of metronidazole, what was the composition of the community when the samples were collected? This is important as the low fiber and control diet may have allowed differential expansion of other members of the microbiota that influenced colonization of the *Bt* strains. Please provide a 16S rRNA gene analysis of the fecal samples at the end of the experiment.

Reply: We thank the reviewer for this insightful comment. We followed the reviewer's suggestions and performed a 16S profiling of the intestinal community at the end of the experiment. Aligned with the reviewer's assumption, the two diets had a significant impact on the composition of the microbiota, as quantified by the taxonomy profiling, beta diversity, and alpha diversity (Extended Data Fig. 1 in the revised manuscript). Notably, the diets did drive the differential expansion of members belonging to *Bacteroidales*, *Clostridiales*, and *Erysipelotrichales* (Extended Data Fig. 1a). This is the very reason we performed the competitive experiment, in which the two experimentally introduced strains experienced the same nutritional environment and interactions with other members of the microbiota in each mouse. This would minimize the skew on strain fitness as a result of altered composition and function of the microbiota as a result of introducing different diets.

14) How did you confirm that all bacteria that grew on the "selective" plates were the particular strain of *B. theta* and not a strain of the microbiota that rebounded?

Reply: In our mouse experiments, we typically plate the fecal contents of antibiotic-treated mice on BHIS agar (in the absence of antibiotic) to ensure that there are no culturable colonies before the inoculation of experimentally introduced *Bacteroides* strains. We also used strain-specific PCR primers to randomly examine colonies we recovered to ensure the strains we introduced were the strains we recovered. Further, the experimentally introduced *Bacteroides* strains lack thymidine kinase (*tdk*), which serves as a negative selection marker for genetic manipulation (PMID: 18611383). In the absence of *tdk*, the experimentally introduced *Bacteroides* strains are resistant to 5-fluoro-2'-

deoxyuridine (FudR). In contrast, members of the microbiota are sensitive to FudR. As such, we supplement our selective plates with FudR to further ensure that we do not recover strains that were not experimentally introduced.

15) Lines 112-113 – remove the phrase “- as opposed to a test tube-” as you previously said the study was performed in vitro and this is unnecessary.

Reply: Thank you for this comment. However, we think that it is important to explain what we mean by the terms “in vitro” and “in vivo” in different contexts. Given that our study starts off with mouse experiments, we initially refer to those as “in vivo”. In the present context, however, “in vivo” refers to bacterial cultures and “in vitro” to cell-free test tubes and we think that by clearly specifying this, we can avoid confusion of our readers.

How was the epitope tagged RbpB created? The first section of the Experimental methods section is titled “Bacterial cultivation and genetics” but there is no mention of how any of the genetic constructs were made. The information contained in Table S2 is not enough. How was the DNA amplified, were the plasmids sequenced to ensure they did not introduce a mutation in surrounding genes.

Reply: This information has now been added to the Methods section (see lines 462-474).

16) There is a Table S2, but no Table S1.

Reply: Supplementary Table 1, to which we refer to in line 137 of our manuscript, contains a list of all the identified RbpB CLIP-seq peaks and their positions in the *B. thetaiotaomicron* genome. We had uploaded it as an EXCEL file along with the other manuscript files and apologize if it had not been passed on to the reviewers.

17) Line 114 – state what the epitope is. State it is a FLAG-tagged version of RbpB. List here if it is N-terminal or C-terminal tag.

Reply: The FLAG-tag was fused to the protein’s C-terminus. This information has now been added to the respective sentence.

18) Was the flag-tagged RbpB that was introduced into the chromosome done so in the delta rbpB mutant? If not, the endogenous RbpB should compete for binding. Or was the tag added to the end of the native gene? Did you show that the flag-tagged RbpB was not impaired for function? You need to show that it can complement the deletion mutant for one of the various functions.

Reply: For our CLIP-seq experiment, the FLAG-tagged *rbpB* allele was inserted into the chromosome of the wild-type background, driven from a constitutive promoter. We are aware of the possibility that the epitope-tagged RbpB thus competes with endogenous RbpB for RNA ligands (note, however, that endogenous *rbpB* was barely expressed under the condition selected for CLIP-seq, so competition with the constitutively expressed RbpB-FLAG is expected to be low).

To provide support for the FLAG-tagged RbpB to be functional, we focused on a molecular phenotype, rather than the mouse phenotype, to minimize animal experimentation. Specifically, we focused on the expression levels of the *susC^{BT_3983}-susC^{BT_3983}* operon as a readout for RbpB functionality. This is because this direct target of the FopS-10 sRNA was downregulated in a delta-*rbpB* mutant relative to wild-type *B. thetaiotaomicron*. In line with *rbpB* being ~12-fold overexpressed in the FLAG-tagged strain compared to the wild-type (see Rebuttal Figure 3A, below),

we found the downregulation of this *susCD* operon in the delta-*rbpB* strain to be overcomplemented in the *rbpB*-flag strain (Rebuttal Fig. 3B).

Rebuttal Figure 3: qRT-PCR analyses of *rbpB* mRNA (A) and of the *susCD* operon of PUL72 (B). Bars denote the mean from six replicate measurements (represented as single dots) and error bars indicate the standard deviation.

19) Line 422 – An OD600 of 2.0 is not mid-log phase. What is the basis of this statement?

Reply: In our hands, in a 50 mL culture volume in an Erlenmeyer flask, *B. thetaiotaomicron* reaches mid-exponential phase after approximately 7 h, which corresponds to an OD600 of 2.0. For reference, please see Figure 1a in PMID: 32678091.

20) What does “200 OD equivalents per condition” mean?

Reply: The culture volume that contains bacterial cells equivalent to 200 optical densities. In the present example, this amounts to 100 mL of the bacterial culture at an OD of 2.0. Of those, 50 mL (100 OD equivalents) were UV-irradiated, while the remaining 50 mL served as non-crosslinked control. We clarify this now in the Methods section (see lines 502-505).

21) Line 424 – what was the volume of bacteria that were irradiated in this 22cm x 22cm plate?

Reply: Please see our answer to the previous comment.

22) The data presented in Fig S1C are not easy to interpret. It seems there should be a way to show the interacting RNAs that are common between the two replicates.

Reply: We now include an MA-plot showing average fold-changes across both replicates, with peaks called as enriched across both replicates in our cross-linked condition highlighted in red (see new Extended Data Fig. 2d).

23) Figure S3 is confusing based on what is written in the legend and text. How does figure S3B show that the “cellular half-lives of the top-enriched mRNA ligands” are reduced when *rbpB* is deleted or overexpressed?

Reply: We agree that these changes were difficult to infer from the previous version of this figure, where we had plotted the transcript levels on a linear scale. In the revised version of this figure (now

entitled 'Extended Data Fig. 3b'), the data was plotted on a log-scale, which better resolves the reduced transcript half-lives when *rbpB* was overexpressed. We further agree that an effect on transcript stability is less pronounced in the *rbpB* deletion mutant and have rephrased the respective text section in the revised manuscript (see lines 150-152).

24) Line 155 – It is stated that “A BLAST search revealed this multicopy sRNA family to be highly conserved within the Bacteroidota (Suppl. Fig. S5a)” yet in this figure you are only showing strains of one family, Bacteroidaceae. You are not even showing other families of the order Bacteroidales. Even within the Bacteroides, RbpB is not conserved as it is absent from *B. ovatus* and *B. fragilis*. The figure legend must also be changed.

Reply: We replaced “Bacteroidota” by “Bacteroidaceae” in the text and changed the figure legend.

25) Line 177 – why does the inverting the 55 nt-long RbpB binding site reduce formation of the higher order complex but not the not the complex itself? Also, the formation of the higher order complex was not abrogated but rather reduced.

Reply: This is a good question and we can only speculate at this moment in time: we currently hypothesize that the lower band (or smear) results from unspecific interaction of RbpB with RNA, whereas the upper, distinct band results from specific binding and derives from a stable RNP complex. RNP formation and structural analysis of RbpB will be the dedicated focus of a follow-up project that we have initiated in our lab. For the time being, we toned down the language when describing these observations.

26) It is stated that “EMSAs also validated that RbpB binds to GibS, again forming higher order complexes (Suppl. Fig. S4b, 180 Suppl. Fig. S6e).” I don't see how Fig S4b shows this finding. In figure S6C, there is less radiolabel of the FopS-10 mutant RNA compared to WT, therefore, as the trend seems the same as in the WT, the conclusions that the mutation “was sufficient to abrogate formation of the higher order RNP complex” is not supported.

Reply: Thank you for pointing this out! We rephrased the text describing these findings in the revised manuscript (see lines 191-194) and enhanced the contrast of the image depicted in Extended Data Fig. 6b to make it better comparable to the corresponding WT EMSA.

27) Lines 211-214 – the data of in support of this in Fig S7C are weak and there is no quantification.

Reply: We have included additional replicates of this experiment and provide the quantification of the northern blot data (now shown in Extended Data Fig. 6g), and toned down the text in the respective Results section describing these findings (see lines 217-221).

28) Line 316 – Bacteroides is not the predominant genus in many individuals. This should be change to “one of the predominant genera....”

Reply: This was changed accordingly.

29) Line 323 – Based on the data shown in Fig S5A, you should not state that RbpB is conserved, or limit the statement to *B. thetaiotaomicron*.

Reply: Thank you for this comment! This statement has been corrected.

30) Line 380 – maybe alter this statement to remove the words “urgently needed”.

Reply: This was removed.

31) Line 382 – not sure the study uncovered a large RNA network controlling PUL translation in Bacteroides”.

Reply: This sentence was toned down.

Reviewer #3 (Remarks to the Author):

In this manuscript, Rüttiger et al studied the regulatory role of the RNA binding protein, RbpB, in the human commensal bacterium *Bacteroides thetaiotaomicron*. Previous work indicated that RbpB binds to RNA in *B. thetaiotaomicron* (PMID: 34251866), however, the identity of the RNA ligands and potential regulatory functions remained unknown. Here, the authors show that RbpB binds to hundreds of transcripts in *B. thetaiotaomicron*, including a group of paralogous small RNAs (sRNAs), called FopS (1-14). Further investigations into the regulatory roles of the FopS sRNAs revealed a conserved sequence motif required for recognition by RbpB and for base-pairing. Surprisingly, FopS and RbpB have opposing functions in regulation of the *susCD* operon involved in carbohydrate utilization. The manuscript is interesting, well-written, and the rationale of the experiments is easy to follow. I have only few comments and suggestions that I listed below.

Reply: We thank the reviewer for this positive evaluation of our work.

Major points:

- Lines 128-129 and Fig. 2d: please provide additional information why the shorter peaks in the CLIP-Seq analysis are associated with mRNAs, whereas longer peaks are linked to sRNAs (is that also true for 5' and 3' UTRs?). Are there several RNA bindings sites on the protein that could indicate the simultaneous interaction of two RNA ligands?

Reply: Thank you for this question! Indeed, longer peaks were also seen for 5' and 3' UTRs (see below, Rebuttal Fig. 4). According to our in vitro data (3-component EMSAs), RbpB may indeed provide binding sites for multiple RNA ligands as we observed a ternary complex consisting of the GibS sRNA and its target mRNA on RbpB (see Extended Data Fig. 9b). In order to test whether this finding may be generalizable to other sRNA-target combinations, we have now repeated the EMSA experiment for two additional characterized *B. thetaiotaomicron* sRNAs—namely MasB (PMID: 38528147) and BatR (PMID: 38294941)—and their respective targets. As shown in new Extended Data Fig. 10c, also these two sRNAs and their respective target mRNAs simultaneously interact with RbpB in vitro, suggesting that the protein might indeed offer multiple binding sites for RNA. Here, we can only speculate that in these ternary complexes, RbpB's interaction with sRNAs may be more extensive (hence the longer CLIP peaks) than its interaction with target mRNAs. However, at this point in time, we do not have experimental data to further support this hypothesis and would therefore leave this open for future investigation.

Rebuttal Figure 4: Peak size distribution in the RbpB CLIP-seq experiment. Plotted are the frequencies for individual peak sizes over all unique peaks (yellow), all sRNA peaks (green), as well as peaks within 5' (light blue) and 3' UTRs (dark blue).

- Lines 189-203, Fig 3: please clarify if all FopS sRNAs have their own promoters or if they are co-expressed with an upstream gene (like FopS-14).

Reply: We now specify in line 216 that FopS-14 is the only 3'-end-derived FopS family member.

- Figs. 4a and c: the selection of fopS deletion mutants that were created seems somewhat random as it remains unclear if FopS 09, 10, 13 are more likely to regulate these target mRNAs, when compared to the other FopS sRNAs. Therefore, it would be useful to create a mutant strain lacking all fopS sRNA genes. Alternatively, the authors could consider to over-express a synthetic antisense RNA that is complementary to all FopS sRNAs and test target regulation.

Reply: Thank you for this comment! Four of the 14 FopS family members of *B. thetaiotaomicron* have insertions or point mutations as compared to the consensus of the 5' FopS sequence (see Fig. 3a). However, for functional assays, we intended to delete individual FopS members whose 5' end perfectly matches the consensus sequence, leaving ten candidates. Among them, FopS-09, FopS-10, and FopS-13 were selected to represent the sequence diversity in the downstream region, between positions 35 and 60. We now mention this explicitly (see lines 231-232).

With respect to your interesting comment about expressing an antisense RNA to sponge FopS sRNAs: we generated a *B. theta* strain that constitutively expresses a 21 nt-long RNA with perfect complementarity to the R1' seed region of 10 of the 14 FopS sRNAs (see below, Rebuttal Fig. 5a). While the respective sponge RNA was highly abundant at all phases of growth (Rebuttal Fig. 5b), neither did it markedly affect FopS steady-state levels (Rebuttal Fig. 5c), nor the mRNA level of the FopS target *BT_3983* (Rebuttal Fig. 5d). Therefore, while we appreciate this comment and will keep trying to establish an antisense sponge strain in the future (e.g. by increasing the length of the antisense region of the sponge RNA), within this revision we did not succeed to inhibit the FopS sRNA family in this manner.

However, we came up with a workaround. That is, we did manage to generate a *fopS* double deletion mutant (devoid of FopS-09 and FopS-10). In this mutant background, we additionally introduced our recently published CRISPR interference system (PMID: 38294941) to transcriptionally silence sRNAs in a targeted manner. In doing so, we managed to reduce FopS levels to ~30%. The respective multi-*fopS* mutant strain showed a strong upregulation of FopS target genes as compared to wild-type bacteria and a growth advantage in liquid medium containing high-mannose *N*-glycans (i.e., the PUL72 substrate) as the sole carbon source (see our response to a comment raised by Reviewer #1:

pages 2-3). These new data are contained in Figure 4d and Extended Data Fig. 8, and the results described in the revised manuscript (lines 264-280).

Rebuttal Figure 5: Efforts to generate an antisense RNA expression strain to knock down FopS. **a)** The anti-FopS sponge RNA was designed in antisense orientation to the highly conserved 5' end (the first 21 nucleotides) of the FopS sRNAs and cloned into the pWW3452 vector (PMID: 28431251) under constitutive induction from a strong phage promoter and in front of a strong ribosomal terminator. The resulting artificial transcript has a combined length of 73 nt (26 nt carryover sequence from plasmid pWW3452 followed by the 21 nt-long anti-FopS sequence, a 6 nt linker, and finally the 20 nt-long terminator). **b)** Northern blot confirms the constitutive expression of the sponge RNA in the Sponge⁺ strain relative to the wild-type (WT) control strain at early-exponential phase (EEP), mid-exponential phase (MEP) and stationary phase of growth in TYG medium. **c)** Expression levels of the FopS sRNAs in the Sponge⁺ strain relative to WT at defined time-points from 4 h to 24 h of growth in TYG medium. 5S rRNA was the loading control. **d)** qRT-PCR analysis of the FopS target *BT_3983* in the Sponge⁺ strain relative to a scrambled control. Since the antisense RNA is complementary also to the GibS sRNA (see panel a), the mRNA levels of two established GibS targets (*BT_0771*, *BT_3893*; PMID: 32678091) were measured in parallel.

- Figs. 4g and f: I think it would be useful to also include point mutations in the base-pairing site of FopS and *susC* as additional controls in these experiments.

Reply: We thank the reviewer for this important comment. For the revised version of this study, we have generated point mutations in the basepairing site of FopS-10 as well as compensatory mutations in the *susC* target region, and analyzed their binding and effect on target translation. The introduced point mutations had to be carefully selected to spare the start codon and the translation-

enhancing adenines at positions -5 to -12 (PMID: 31602466) of the *susC* mRNA. We tested various mutations and eventually identified an 8-nt exchange that was sufficiently harsh to completely abolish in vitro interaction with FopS-10 as judged from EMSA experiments, while still being efficiently translated. Importantly, introduction of the corresponding compensatory mutations within the FopS-10 seed region fully restored binding (see new Extended Data Fig. 7d) and target repression (new Extended Data Fig. 10b). We therefore believe that we have now convincingly shown that FopS-10 represses *susC*^{BT_3983} translation in a direct and sequence-specific manner.

Minor points:

- Line 125: please provide additional information of the 14 intergenic peaks identified in the CLIP-Seq analysis. Are these additional, previously unidentified, sRNAs?

Reply: Thank you for this comment! Cross-comparison with our previously published transcriptomic datasets (PMID: 32678091; 38528147) suggested that at least CLIP peak#29 might indeed derive from a new sRNA. This is because there was some read coverage at this genomic location (especially, when *B. theta* was grown in deoxycholate or simple sugars including glucose, arabinose or xylose), yet in the absence of a clear transcription start site, we had not previously annotated it as an sRNA. In the case of CLIP peaks #199 and #200, while originally falling within intergenic regions, according to the latest release of the corresponding RefSeq annotation, these peaks fall within ribosomal RNA genes (*BT_r14* and *BT_r15*). To document all the above, a respective comment has been added to CLIP peaks #29, #199, and #200 in Supplementary Table 1. The positions of the 11 remaining intergenic peaks did not accumulate any reads in our previous transcriptome atlas, implying that, if they would correspond to hidden sRNAs, these sRNAs might be generally of very low abundance. By this occasion, we also noticed that the original manuscript had not always reported the exact peak numbers, which we have now corrected in the revised version.

- Line 145: the 41 nt-long motif is somewhat confusing as the binding peaks for mRNAs (the majority of interaction partners) are only ~30nt long.

Reply: The 41 nt-long motif depicted in Fig. 2e is contained exclusively in sRNAs, whose binding peak lengths range between 50 and 100 nt (green in Fig. 2d).

- Line 180: I think it would be useful to provide additional information on GibS at this point (or in the introduction). Otherwise, this paragraph is difficult to follow.

Reply: We introduce GibS in the paragraph above (lines 178-181).

- Paralogous sRNAs are often regulated by a universal sponge sRNA. Is there evidence for such a sponge sRNA in *B. thetaiotaomicron*?

Reply: We initially discovered the FopS paralogs by performing a BLAST analysis of the R1' sequence in the *B. thetaiotaomicron* genome. An extended BLAST analysis against the entire RefSeq database identified different *Bacteroidaceae* species with varying numbers of hits (i.e., FopS homologs) that matched that same sequence. When running BLAST searches on the reverse-complemented R1' sequence, the same hits came up. However, none of the reverse R1' DNA sequences seems to be transcribed, as a BLAST against the RefSeq_rna database did not result in any hits. An Rfam search likewise failed to identify any matches to this sequence. We therefore conclude that, if such an

endogenous FopS sponge exists, it was either not abundantly expressed in previous experiments and/or possesses a somewhat degenerate sequence, yet not the perfect reverse-complement to R1'.